# HESS Opinions: Are soils overrated in hydrology?

Hongkai Gao [1,2]*, Fabrizio Fenicia [3], Hubert H.G. Savenije [4]

[1] Key Laboratory of Geographic Information Science (Ministry of Education of China), East China Normal University, Shanghai, China

[2] State Key Laboratory of Tibetan Plateau Earth System and Resources Environment (TPESRE), Institute of Tibetan Plateau Research, Chinese Academy of Sciences, Beijing, China

[3] Eawag, Swiss Federal Institute of Aquatic Science and Technology, Dubendorf, Switzerland

[4] Water Resources Section, Delft University of Technology, Delft, the Netherlands

*Corresponding to: Hongkai Gao (hkgao@geo.ecnu.edu.cn, gaohongkai2005@126.com)

## Abstract

Traditional hydrological theories are based on the assumption that soil is key in determining water's fate in the hydrological cycle. According to these theories, soil hydraulic properties determine water movement in both saturated and unsaturated zones, described by matrix flow formulas such as the Darcy-Richards equations. They also determine plant available moisture and thereby control transpiration. Here we argue that these theories are founded on a wrong assumption. Instead, we advocate the reverse: the terrestrial ecosystem manipulates the soil to satisfy specific water management strategies, which are primarily controlled by its reaction to climatic drivers and by prescribed boundary conditions such as topography and lithology. According to this assumption, soil hydraulic properties are an "effect", rather than a "cause" of water movement. We further argue that the integrated hydrological behaviour of an ecosystem can be inferred from considerations about ecosystem survival and growth, without relying on internal process descriptions. An important and favourable consequence of this climate and ecosystem-driven approach is that it provides a physical justification for catchment models that do not rely on soil information and on the complexity associated to the description of soil water dynamics. Another consequence is that modelling water movement in the soil, if required, can benefit from

the constraints that are imposed by the embedding ecosystem. Here we illustrate our ecosystem
perspective of hydrological processes and the arguments that support it. We suggest that advancing
our understanding of ecosystem water management strategies is key to building more realistic
hydrological theories and catchment models that are predictive in the context of environmental
change.

## 1    A change in perspective

Soil is important in hydrology. Soil forms the substrate of the terrestrial ecosystem and hence it is
a crucial element of the critical zone of life on Earth (Lin et al., 2006; Banwart et al., 2017).
Through its porous structure, exercising capillarity against gravity, it provides water storage
against droughts and nutrients for plant growth.
It has been argued that the soil forms an ecosystem in itself, full of micro-biotic and macrobiotic
life (Ponge, 2015; Weil and Brady, 2017). Fungi forming dense underground networks live in
symbiosis with vegetation, exchanging nutrients for carbon, which makes them responsible for the
larger part of subterranean carbon storage (Domeignoz-Horta et al., 2021). Soils are full of life.
Above ground life cannot survive without sub-surface life; they are part of the same ecosystem.
Soils are embedded in the terrestrial ecosystems, which through evolution and natural selection,
have found ways to make best use of its resources. The processes and structure of a terrestrial
ecosystem are mainly controlled by external factors which are largely prescribed. Among them,
climate plays a major role, as rainfall patterns and seasonal temperatures strongly affect the
distribution of vegetation types; other external factors include topography, lithology, which
determines parental material, and potential biota (Chapin et al., 2011). Given these boundary
conditions, a terrestrial ecosystem adjust its internal behaviour to satisfy its needs, and it
manipulates the substrate on which it grows.
In particular, the soil is the result of a long-term evolution of terrestrial ecosystems given their
boundary conditions. The classic *clorpt* model presented by Hans Jenny's famous 1941 book "The
Factors of Soil Formation" states that $s = f(cl, o, r, p, t, ...)$, where soil properties ($s$) are seen as
a function of climate (*cl*), biotic effects (*o* for organisms), topography (*r* for relief), parent material
($p$), time ($t$), additional factors such as fire (represented by the dots) (Huggett, 2023). This model
suggests that soil properties are largely determined by the embedding ecosystems.
Managing water is an essential task of terrestrial ecosystems, as water is essential to life. And it is
not a trivial task, as it implies bridging dry weather periods, but also avoiding troubles caused by
sustained or heavy rainfall, such as water stagnation or soil erosion. We argue that terrestrial
ecosystems achieve this balance by manipulating key hydrological characteristics such as
interception capacity, infiltration capacity, moisture storage capacity, preferential pathways to
replenish moisture stocks and recharge, and subsurface drainage. According to this view, a
terrestrial ecosystem manipulates the soil hydraulic properties to satisfy specific water
management strategies.
Yet, the most established hydrological theories parameterize water fluxes using soil attributes such
as texture, porosity, moisture retention capacity, wilting point, plant available moisture, etc. (e.g.
Drewniak, 2019, Lu et al., 2019). These theories assume that soil properties are controlling
processes such as infiltration, drainage, or plant evaporation. But this is the wrong way round. Soil
properties are the effect, rather than the cause of water movement, which itself, is governed by the
behaviour of the embedding terrestrial ecosystem.
We therefore argue in favour of an ecosystem-based approach where the integrated hydrological
behaviour of an ecosystem is inferred based on its water management strategies needed to survive
and grow, without relying on internal process descriptions. As we shall see, this is not a
prohibitive task. The very existence of an ecosystem already provides many indications about its
ability to manage its water resources.
This ecosystem-based approach has several beneficial consequences for hydrology. First, it
provides a physical justification for the development of catchment scale hydrological models that
directly rely on the external factors that influence terrestrial ecosystems, such as climate,
topography, lithology, etc. These models would be more realistic than soil-based models because
based on the correct cause-effect relationships. Moreover, they would be less data demanding and
simpler, as they would not require soil texture information and detailed description of soil water
dynamics. Second, it would allow digging into the small scale, if this is deemed necessary,
exploiting the constraints that are imposed by the behaviour of the larger scale system.
In the following, we first present the soil-centred hydrological perspective and its limitations
(Section 2), we then argue that there is limited evidence that soil properties actually matter in
catchment hydrology (Section 3), next we illustrate our terrestrial ecosystem perspective (Section
4), and provide an interpretation of why the soil-based modelling tradition has proliferated in
hydrology (Section 5), finally we illustrate the limitations of our approach (Section 6), and present
our conclusions (Section 7).

## 2    Limitations in the soil-centred hydrological perspective

### 2.1    Challenges in small-scale theories of soil water dynamics

It is a deeply rooted perception in hydrology that small-scale soil water dynamics are key in
determining the integrated catchment behaviour at larger scales such as the partitioning of rainfall
between evaporation, drainage and storage (Vereecken et al., 2022). For example, soil is assumed
to control plant evaporation, as plant available water content is often parameterized as a function
of soil texture (Yang et al., 2016). Processes such as Hortonian overland flow, saturation excess
overland flow, or percolation are often described in relation to water movement in the unsaturated
zone, using laboratory-scale matrix flow theory developed by soil physicists. This theory describes
flow in porous media based on equations that depend on soil hydraulic properties (e.g. porosity,
hydraulic conductivity). Darcy's law describes matrix flow under saturated conditions through a
porous medium under a head gradient. Richards' equation regards matrix flow under unsaturated
conditions in the vadose zone, determining water flow direction and velocity. Numerous
simplified semi-empirical soil infiltration equations were also derived to simulate the infiltration
excess overland flow, such as the Philip and Horton equations (Schoener et al., 2021). The matrix
flow theory is regarded as well-established, much like classical mechanics.
For a hydrological model to be considered "physically-based", it is generally assumed that it needs
to be based on these small-scale theories. Land surface models (LSMs) are strongly based on these
matrix flow equations (Freeze and Harlan, 1969; Lawrence et al., 2019), which determine soil
water movement vertically and laterally (Duffy, 1996; Refsgaard et al., 2022). Even the
representative elementary watershed (REW) approach (Reggiani et al., 1998), a physically based
framework that describes catchment scale processes, is based on the integration of small-scale
conservation equations developed for porous media.
This soil-centred perspective is highly rated in the hydrological community. Some of the most
prestigious hydrology awards exemplify the tribute of the hydrological community to this
perspective, such as the Henry Darcy medal of hydrological sciences in the European Geosciences
Union (EGU), the Robert Horton American Geophysical Union (AGU) hydrological science
medal, which are named after two hydrologists that pioneered the soil-centred approach.
Tracer field experiments, such as dye and isotope studies, have shown that matrix flow is rarely
observed. Most soils contain crevices, preferential channels, and openings that transmit free water
quite rapidly to the sub-surface, which is termed preferential flow (Beven and Germann, 2013;
McDonnell et al., 2007; Beven, 2018; Zehe et al., 2021). Hence, natural conditions do not
resemble well-prepared homogenous soil that can be recreated in a laboratory.
In response to this criticism, soil-water theories have become more complex, allowing for
preferential flow, which required even more detailed soil characterizations. These challenges have
stimulated the development of dual-continuum, dual-porosity, or dual-permeability modifications
(Jarvis et al., 2016), most models are still based on matrix flow theory (Weiler, 2017). Because of
the extreme complexity of soil preferential flow in nature, it is extremely hard to develop accurate
models that describe it, even at the plot scale. The challenge is exponentially greater when
upscaling preferential flow from plot-scale to hillslope or catchment scales (Davies et al., 2013;
Germann, 2014; Or, 2020). At the global scale, hyper-resolution land surface models, which are
deemed necessary to addressing critical water cycle science questions and applications, can have
up to $10^9$ unknowns (Wood et al, 2011)!
From its establishment, preferential flow theory was regarded as the main culprit challenging the
foundation of "physically-based" hydrological models. This avenue has led to models that require
many space and soil-dependent parameters that are difficult to measure, that require massive
computational resources, and that when calibrated are prone to equifinality. Arguably, the avenue
of building more complex models by increasingly detailed representation of soil water movement
is a steep one. But is it a necessary one, if the objective is to build a physically-based model of
catchment scale hydrological processes?

## 2.2 Limitations in the pedotransfer functions approach

Soil-centred bottom-up hydrological models rely on estimates of soil hydraulic properties (SHPs),
such as water retention characteristics and unsaturated and saturated hydraulic conductivity. As
these properties are difficult to measure at appropriate scales, soil pedotransfer functions (PTFs)
have been developed to express SHPs as a function of more accessible soil properties, such as soil
texture (i.e. sand-, silt-, clay- content), organic matter, and bulk density (Figure 1; van Looy et al.,
2017; Or, 2020; Haghverdi et al., 2020; Gupta et al., 2021; Hohenbrink et al., 2023).

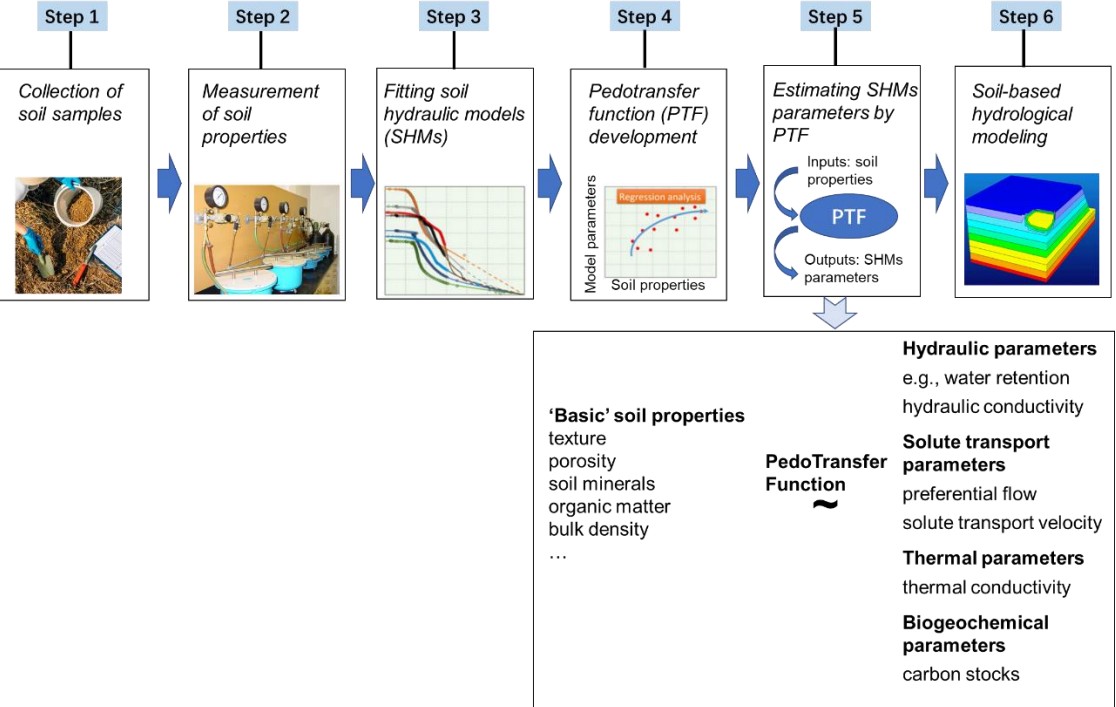


Figure 1. Schematic illustration of data collection, laboratory measurements, fitting soil hydraulic
models (SHMs), pedotransfer function (PTF) development, and soil-based hydrological modelling
workflow (adapted from Van Looy et al., 2017 and Haghverdi et al., 2020).
There are several critical issues with the practicality and accuracy of this approach: 1) most soil
property parameters are measured by pedologic surveys, at great expense and efforts (Van Looy et
al., 2017). 2) PTFs are usually obtained by using measurements from uniform soil samples, and
performed in laboratory-scale experiments, which merely reflect disturbed and therefore unnatural
conditions; 3) the parameters obtained at the laboratory scale are not necessarily the same as at the
model scale, which requires upscaling assumptions, which are difficult to verify or recalibrate,
hampered by equifinality.
Unfortunately, readily-available soil information (e.g., texture, bulk density, organic matter)
correlates poorly with soil hydraulic properties. Gutmann and Small (2007) have shown that soil
textural classes, across a range of climates and vegetation covers, merely explained 5% of the
variance of real SHPs. In another study, it was found that 95% of the default soil hydraulic
parameters in a state-of-the-art land surface model, largely based on soil textural data, were
significantly different from region-specific observations (Kishné et al., 2017).
Recent studies showed that in order to achieve more realistic estimates of soil hydraulic properties
it is necessary to include information about vegetation or biophysical activity (Or, 2020). For
example, Bonetti et al. (2021) proposed soil structure corrections into pedotransfer functions,
informed by remote-sensing vegetation metrics and local soil texture. Additional studies
"rebalance" the soil texture information and highlight the importance of soil structure, originated
by soil biophysical activity (Or, 2020; Fatichi et al., 2020). Not only the "physically-based"
models, but also the empirical soil-based models, for example the soil conservation service (SCS)
method in the SWAT model (soil water assessment tool), involve land-use data to "rebalance" the
soil-based curve number in catchment simulations (Arnold et al., 2012).
Building realistic pedotransfer functions requires detailed characterization of the soil, requiring a
large number of parameters that are difficult to estimate. This approach, while feasible for a
hillslope or a headwater catchment, becomes impractical at regional or global scales. For
hydrological purposes, the ultimate goal is often to determine integrated fluxes of hydrological
response at large scales. Hence, it is worth asking: can this integrated behaviour be determined
directly from observations, without resorting to small-scale theories and upscaling assumptions?

## 3 Does soil variability matter in catchment hydrology?

### 3.1 Do soil-centred models reproduce hydrological variability?

A key objective of "physically based" models is to represent hydrological variability, such as spatial patterns of soil moisture, runoff or evaporation. Figure 2 is a revealing illustration of how a "physically based" model that relies on detailed soil information can make inconsistent predictions under extreme circumstances (Beekman et al., 2014). On average, these models may function adequately, as almost all hydrological models do under average conditions, but the example of Figure 2 shows how during a relatively extreme drought in The Netherlands the modelled evaporation is unrealistic.

The top panels in Figure 2 show remote sensing derived evaporation obtained by interpolation of eddy-covariance and lysimeter observations using ETLook, an energy balance-based evaporation product (Bastiaanssen et al., 2012). The bottom panels in Figure 2 show evaporation modelled with the Netherlands Hydrological Instrument (NHI) distributed model, which heavily relies on detailed soil data. The two methods for estimating evaporation are independent, and arguably, the ETLook approach is more realistic, as it is based on eddy-covariance observations. The comparison is presented for two dry summer months in 2006: June (left panel) and July (right panel).

Two aspects of this comparison are striking. First, in terms of temporal dynamics, ETLook evaporation estimates show an increase in response to increased evaporative demand, whereas the NHI evaporation estimates are decreasing, in response to water stress. Second, in terms of spatial patterns, ETLook estimates are more uniform in response to relatively uniform climatic conditions, whereas NHI estimates are highly variable, mimicking the variability of the soil maps used in the model, which are used to determine plant available storage. The July 2006 picture in the bottom panel, in fact, mimics the soil map. Red (high evaporation) is seen on clay soils and purple (almost no evaporation) on sand.

It is interesting to observe that according to ETLook (top right) the forested sandy part at the centre of The Netherlands was evaporating lushly, whereas according to the hydrological model

(bottom right), this ecosystem appeared to be dead. Apparently, the ecosystems continued to
evaporate well during July 2006, in spite of the dry weather conditions. Our interpretation is that
the ecosystems had prepared for this eventuality and had created enough rootzone buffer to
overcome this period of drought, compensating for the variability of soils.
Although such mismatch between distributed model outputs and remote sensing monitored
patterns are not infrequent, they are typically not regarded as a challenge to the basic model
assumptions, but rather, as a problem associated to the uncertainty in model inputs. Hence, such
soil-centred hydrological models remain vivid under the hope that "novel, highly resolved soil
information at higher resolutions than the grid scale of LSMs may help in better quantifying sub-
grid variability of key infiltration parameters" (Vereecken et al., 2022). But is this a realizable
hope?

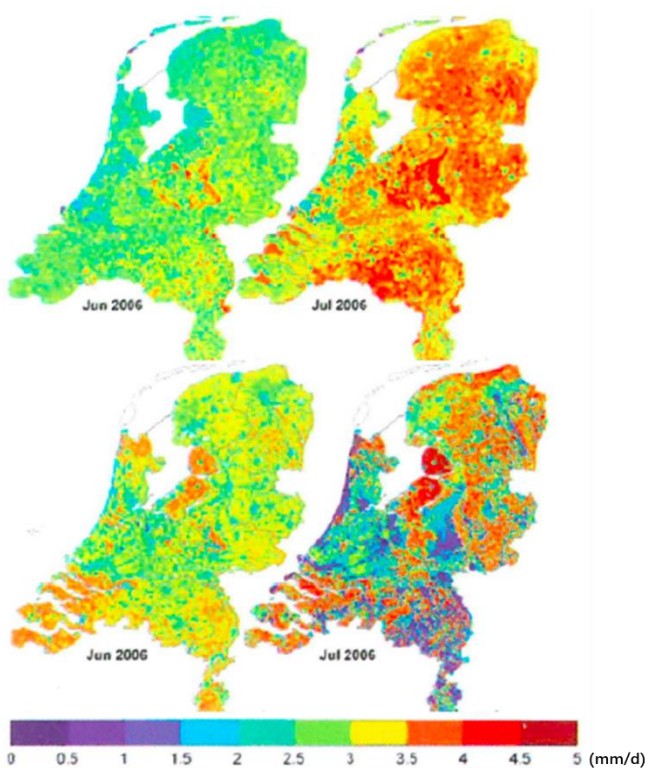


Figure 2. Evaporation during June (left) and July (right) of 2006 in The Netherlands. Remote
sensing derived above, modelled with a "physically-based" hydrological model below (from
Beekman et al., 2014)

## 3.2 Is soil a good predictor for streamflow spatial variability?

The top-down approach is a common way to infer internal catchment behaviour and its controlling factors from catchment response data (Sivapalan et al., 2003). For example, this approach has often been used to interpret spatial variability of streamflow based on controlling factors such as climate, vegetation, topography, geology and soils. Interestingly, in these applications, soil properties are often a poor predictor of streamflow variability. For example, Addor et al. (2018) used 671 catchments in the USA and found that, compared to soil properties, landscape features, i.e. vegetation and topography, have stronger correlations with hydrologic signatures, not only for average streamflow, but also for high-flow, low-flow, and streamflow seasonality (Figure 3).

One the arguments in favour of high resolution distributed models has been their ability of spatial extrapolation, such as capturing the spatial variability of streamflow. Such extrapolation ability cannot be achieved by lumped models that rely on calibration on each individual catchment. However, there are now several examples of catchment scale distributed models that describe the spatial variability of streamflow without relying on soil information (e.g. De Boer-Euser et al., 2016; Fenicia et al., 2016; Gao et al., 2019; Dal Molin et al., 2020; Fenicia et al., 2022). These models are clearly more complex than lumped models, but not orders of magnitude more complex, as they distribute parameters according to a small number of landscape units.

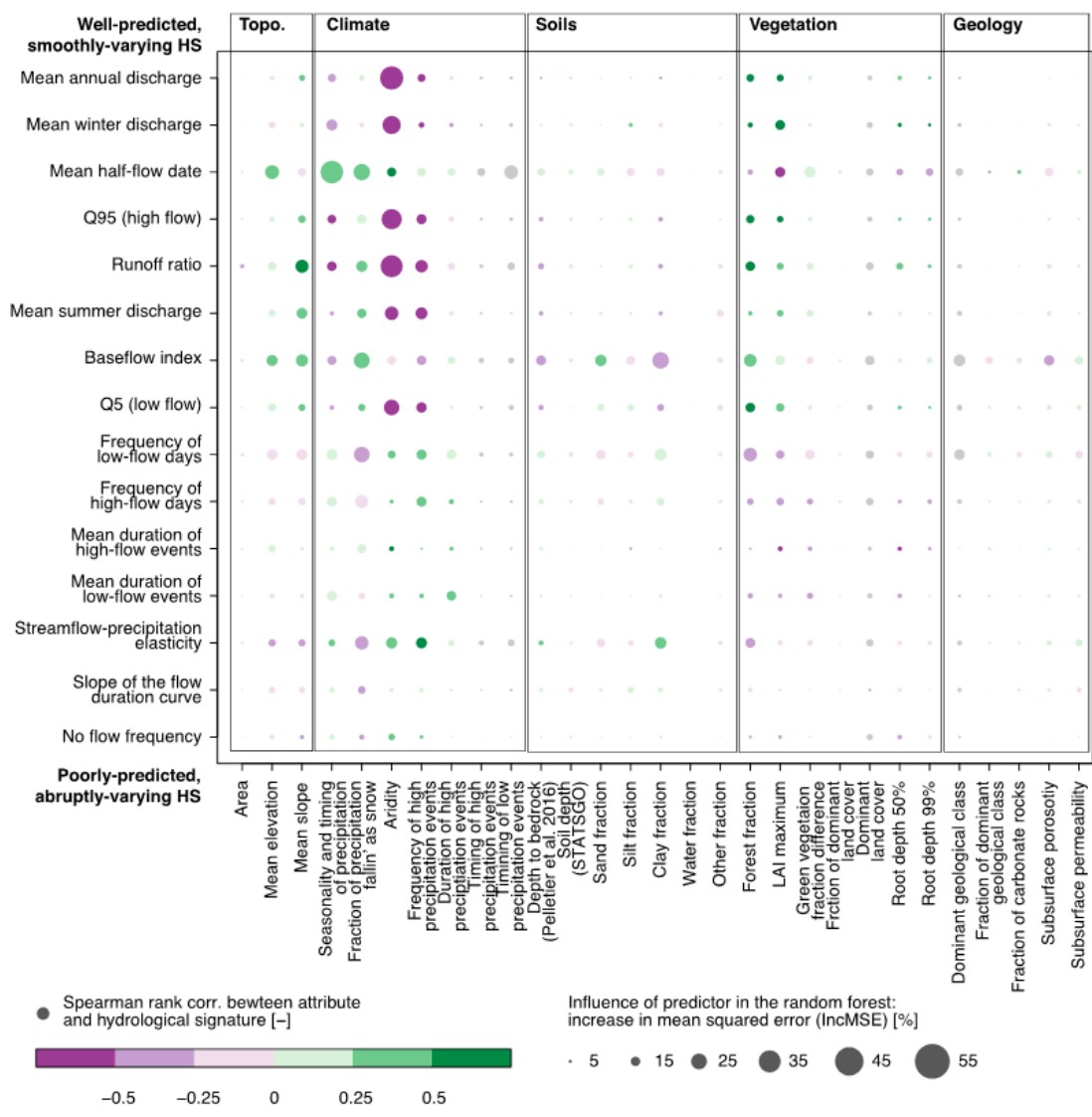

238

Figure 3. Comparison of the influence of catchments attributes and hydrological signatures for 671 U.S. watersheds (from Addor et al., 2018). Large, brightly coloured circles imply strong correlations and high influence. (Reprinted by permission of John Wiley and Sons).

## 4  Putting the terrestrial ecosystem at the centre of hydrology

### 4.1  Ecosystem hierarchy

It has been shown that terrestrial ecosystems largely respond to external climate forcing, and to the lower boundary conditions determined by topography and lithology (Chapin et al, 2011). With time, terrestrial ecosystems organize themselves to make best use of the available solar energy and resources. Hence, they adapt to the climate, by developing vegetation types in response to rainfall

patterns and seasonal temperatures. They also develop the soil given the climate, organisms,
topography and parental material as suggested by the *clorpt* model (see Section 1).
Our view, consistent with this perspective, is that an ecosystem adjusts the soil hydraulic
properties to fulfil specific water management criteria. Hence, understanding the water
management strategies of the ecosystem is a prerequisite to understanding and modelling soil
processes. This perspective is opposite to the classical soil-centred hydrological perspective
presented in Section 2, which sees water fluxes, such as evaporation and drainage, as a function of
soil properties.
Figure 4 further illustrates our ecosystem view and how it differs from the classical approach in
hydrological science. The traditional view is represented by the four isolated circles in the left
panel of Figure 4. This view assumes that soil plays a central role in governing the terrestrial water
cycle. In particular, depending on climate forcing, soil hydraulic properties will determine water
availability for vegetation and water fluxes such as percolation and surface runoff. According to
this view, the understanding of soil water processes is a prerequisite to simulate vegetation
dynamics and water fluxes. The circles are isolated from each other, reflecting that in this view
soil properties, vegetation cover and climate are seen as independent on each other, and can
influence independently hydrological processes. Indeed, hydrological models typically
parameterize soil and vegetation independently from each other and from climate forcing.
Our view is represented by the nested circles in the right panel of Figure 4. Climate sets the
boundaries for the terrestrial ecosystem, and in turn, the ecosystem manages its water resources,
determining hydrological processes. Soil hydraulic properties are a function of the ecosystem
water management strategies. The circles are nested to reflect the hierarchy between them, in the
sense that internal circles are dependent on the external ones. The double arrows indicate that
there are feedbacks between these circles, but the influence of the external circles on the internal
ones is much greater than vice versa. More specifically, local climate has a strong effect on an
individual ecosystem, which intentionally adapts to it, developing strategies to grow, survive and
reproduce. In turn, an individual ecosystem cannot change the local climate significantly
according to its needs. Hence, the feedback of an ecosystem on the climate is smaller and less
"intentional" than the effect that the climate exerts on an ecosystem. Similarly, the control that the
terrestrial ecosystem exerts on soil hydraulic properties, mediated by its water management
strategies, is much greater and purposeful than the control of the soil on the embedding ecosystem.
In our perspective, such hierarchy and interactions can reduce rather than add complexity and
facilitate hydrological process understanding and modelling. For example, it provides a
justification for the level of detail of catchment models. In many applications of catchment
hydrology, the 'ecosystem circle' represents the necessary level of detail, and as the effect of soil
on the ecosystem is rather minor, it is unnecessary to dig into what happens within the 'soil water
circle'.

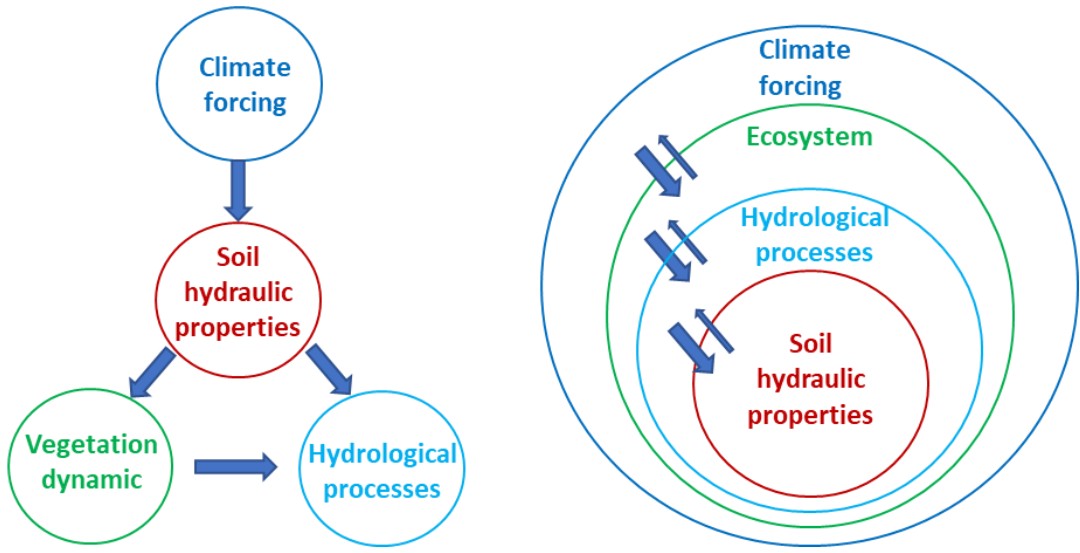


Figure 4. The isolated circles (left) represent the traditional soil-centred hydrological perspective.
The nested circles (right) represent our view of ecosystem hierarchy and cause-effect
relationships.
**4.2   The ecosystem is the ultimate water manager**
An ecosystem that results from a process of evolution contains traits that are functional to its
survival. In this perspective, it is important to understand what the system is trying to achieve in
order to explain and predict its behaviour. In the context of hydrology, this approach requires to
understand (i) which water management strategies the ecosystem needs to adopt in order to sustain
itself and survive, (ii) how hydrological processes, such as interception, surface runoff (or lack
thereof), subsurface stormflow, contribute to satisfy the water needs of the ecosystem, and (iii)

which physical characteristics the system needs to demonstrate to enable such processes. This evolutionary perspective considers the structure and internal processes of the ecosystem dependent on its overall behaviour, and it is contrary to the static approach which underlies typical description of soil hydrology, where the system structure is seen as prescribed, and small-scale processes are assumed to determine overall system behaviour.

So, what are the water strategies of the ecosystem, and how do they affect its structure and internal hydrological processes? Humans are well aware that water management is critical to their survival. For this reason, they have developed activities for optimum use of water resources such as flood control, water storage, water conservation, river regulation, irrigation, and water treatment. Similarly, a natural ecosystem can only survive if it organizes its water resilience. In other words, if an ecosystem had not organized its water resilience, it would not have survived and would no longer exist. The very existence of an ecosystem tells us several aspects of its water management strategies. In particular:

- An ecosystem needs to provide sufficient moisture storage in the rootzone, so that vegetation can overcome critical dry spells, but also sufficient infiltration capacity and subsurface drainage to maintain moisture levels between acceptable boundaries: not too wet and not too dry.

- Runoff ,the excess water after precipitation has replenished the ecosystem's water deficit needs to be drained as quickly and efficiently as possible.

- Preventing surface runoff is an essential need of an ecosystem, which serves to avoid soil erosion. Indeed, surface runoff is seldom observed on vegetated hillslopes. It does occur on bare rocks, where there is no vegetation, or on floodplains, where saturation overland flow does not cause significant erosion. Also, it occurs in disturbed ecosystems, such as urbanised areas, roads, paths and ploughed agricultural fields. In rare cases, such as the Loess Plateau in China, the failure of surface runoff prevention caused severe soil erosion at local scale and disastrous sediment deposition and flooding in the lower Yellow River.

- The ecosystem needs to retain nutrients and soil particles and to retain water for plants.

For this reason, it creates preferential flow paths that facilitate infiltration while retaining

moisture and nutrients in retention zones. If there is too much water, then excess water

bypasses the rootzone where it can recharge the groundwater or is evacuated through

preferential sub-surface drainage patterns on hillslopes. This type of drainage generates

subsurface storm flow and recharges the groundwater system.

•    Ecosystems will generally avoid catastrophic events such as death from drought,

temperature stress, landslides, windthrows or fires. If such disruptive events occur, it is

generally at time scales longer than ecosystem memory. If disturbances occur more

frequently, ecosystem generally develop resilience to them, such as in the case of frequent

fire, where ecosystems can develop fire resistant species, or vegetation that can recover

biomass more quickly (Chapin et al., 2011).

Considering hydrological processes in the context of their purpose from an ecosystem perspective
can clarify cause-effect relationships and therefore help their conceptualization and modelling. For
example, it can constrain plausible values of SHPs, which can be determined based on
considerations about overall system behaviour.

## 4.3   The rootzone is the key element in hydrology

From a catchment hydrology perspective, a key objective is to determine the partitioning of
precipitation between evaporation, drainage and storage. This partitioning mostly takes place in
the rootzone. The vertical profile of the critical zone can be divided into different layers, i.e.:
canopy, litter layer, rootzone, water transition zone, unconfined groundwater, and confined
groundwater. The most significant phase change of water happens in the canopy, litter layer, and
rootzone. Once water overtakes these zones, evaporation is relatively small and water is routed to
the stream through various pathways. Globally, the vegetation interception storage capacity of
terrestrial ecosystems is about 1-2 mm, as estimated by remote sensing-based LAI data (De Roo et
al., 1996). The litter layer storage capacity differs among ecosystems, but it is likely to increase
the total interception storage capacity to around 2-5 mm (Shi et al., 2004; Gerrits et al., 2010).
Global average rootzone storage capacity in vegetated regions is about 146-242 mm, as estimated
by multiple approaches and datasets (Kleidon, 2004; Wang-Erlandsson et al., 2016), which is
significantly larger than interception and litter layer water storage capacities. Therefore, the
rootzone storage is the one with the longest memory, which influence how much precipitation
eventually becomes streamflow.
Referring to common hydrological models, the rootzone storage can be assimilated to the
"production" reservoir in the GR4J model (Perrin et al., 2003), the "upper zone" reservoir in HBV
(Lindström et al., 1997), the "tension water" storage in Xinanjiang model (Zhao, 1992), or the
"soil moisture" storage in probability distributed model (PDM) (Moore, 2007). In these models,
the size of this reservoir is typically obtained by calibration. This approach is clearly
unsatisfactory from a theoretical point of view as it makes these models not predictive under
environmental change.
From a soil-based perspective, the rootzone storage is commonly estimated as a function of plant
available moisture and rooting depth (Yang et al., 2016). In our view, this approach is also not
satisfactory, as it considers plant available moisture and rooting depth as independent variables,
and rootzone storage as the dependent variable. We argue the reverse: plant available moisture and
rooting depth are a function of the rootzone storage that is created by the ecosystem to fulfil its
water management strategies. Moreover, the classical approach is impractical, as obtaining the
detailed spatio-temporal root and soil information at a global scale is virtually impossible (Or,

2020).

So, how to determine rootzone storage without resorting to calibration, or in situ measurements?
As mentioned in the previous section, our ecosystem approach would start with understanding the
ecosystem water management strategies, and using this understanding to figure out how the
ecosystem needs to organize its internal behaviour. Vegetation will try to maintain evaporation
close to potential to maximise net carbon profit. It will therefore optimize its rootzone water
storage so that it is sufficiently large to overcome typical dry spells, much like humans size dams
to sustain droughts (Gao et al., 2014). An approach that appeared to work well locally and globally
for estimating the rootzone storage capacity is the mass curve technique, originally developed for
reservoir design at an acceptable probability of failure (Gao et al., 2014; Wang-Erlandsson et al.,
2016). Here the supply is represented by precipitation, and the demand by potential evaporation.
This technique is uniquely based on climate data. This technique has an important benefit over
approaches based on calibration or field observations: it can also be used to describe how the
rootzone would evolve in response to climate change.
It is worthwhile noting that rootzone and soil have a strong connection but are essentially different
things. The soil profile can reach over hundreds of meters depth, e.g. the Loess Plateau in China
(Zhang et al., 2014), of which only the rootzone is the active area, whereby the soil is merely the
substrate of it. Rootzone storage can also be larger than soil water storage, for example in karst
mountainous areas where soil is thin and discontinuous, bedrock storage serves as an important
source of plant-available water (McCormick et al., 2021). In very dry climates, roots can even
reach the deep groundwater, thus in this case, the rootzone also includes some part of the
groundwater (see Singh et al., 2020). In cold regions, it is necessary to take account of snowmelt
and soil freeze/thaw processes on rootzone water storage and resulting hydrologic connectivity
(Gao et al., 2020; 2022). In cropland, where irrigation provides an extra water supply to rootzone
during dry seasons, the rootzone water storage capacity is often smaller than under natural
conditions with similar climate conditions (Xi et al., 2021).

## 4.4  Landscape-based model: the giant view of hydrology

A soil-based model of catchment scale processes is like the ant's perspective, observing a complex
world of heterogeneities and randomness (Savenije, 2010). According to this perspective, small-
scale processes are the basis for integrated system behaviour. As a result, a model can be
"physically-based" only if it relies on small scale physics.
Seeing the patterns of hillslope, landscape and catchment is rather the giant's perspective, as these
patterns only become visible when we zoom out well beyond the microscale of the soil or the
human scale (Savenije, 2010; Gao et al., 2018). Landscape, as the integration of topography and
landcover, is seen as the long-term co-evolution of ecosystem, atmosphere, lithosphere,
pedosphere, hydrosphere, and human activities (Wu, 2013; Troch et al., 2015). According to this
perspective, a "physically-based" model needs to be based on large-scale system behaviour.
Both approaches can produce models that provide good results. However, in our perspective, for
catchment hydrology applications it is the giant's perspective that wins. First, the giant's model
captures the right cause-effect relationships, and is therefore more satisfactory from a theoretical
point of view. For example, it is a tool to test how an ecosystem would adapt to changes in
climatic drivers. Second, landscape-based catchment models will generally be simpler than
fragmented catchment models, as natural system exhibits emergent properties, which effectively
enable a description of large-scale processes independent on what happens at the smaller scale.
Such emergent properties are often characterized by simple laws, such as the fill and spill bucket
model with thresholds and associated time scales (McDonnell et al., 2021), and the linear reservoir
for groundwater at hillslope and catchment scale (Savenije, 2010; Fenicia et al., 2011; Savenije
and Hrachowitz, 2017). Interestingly, the groundwater system also appears to be self-organized
and structured (Savenije, 2018). For example, the recession parameter $k$ is around 45 days in
worldwide catchments regardless of their climate, topography, soil, and geology (Brutsaert, 2008).
Discovering these properties and related signatures benefit our understanding and prediction of the
dynamic adaption of ecosystems to environmental change, and the subsequent impacts on
hydrology (Gharari et al., 2014; Jackisch et al., 2021).
This ecosystem perspective provides a physical justification for catchment scale models that do
not rely on small scale physics, as they are independent on what happens at the smaller scale.
Moreover, they can provide a constraint to smaller scale processes, and therefore facilitate their
representation. For example, the partitioning of water between evaporation, storage and release
that characterize the larger scale system can be used to constrain plausible values of difficult to
measure soil properties such as rooting depth, plant available moisture and hydraulic conductivity.
This can favour more accurate descriptions of soil water dynamics, which, although often
unnecessary for typical catchment scale applications, can may be important for other purposes.

### 4.5   Proposed modelling steps in poorly gauged catchments

How can this approach be implemented in modelling an ungauged catchment? There are the
following steps to be considered as a quick guide to model building.
The first thing is to classify the basin on landscape and geology. This determines model structure.
It defines the proportion between the major three fast runoff mechanisms: rapid subsurface flow
(for Hillslope), saturation overland flow (for Wetland) and Hortonian overland flow (for Plateau,
bare rock). Theory and application of landscape-based modelling are presented in (Savenije, 2010;
Gharari et al., 2011, Fenicia et al., 2011; Gao et al., 2014, 2018; De Boer-Euser et al., 2016;
Hulsman et al., 2021a; Bouaziz et al.,2022).
Subsequently classify landscape units on ecosystem, land use and climate. The climate and the
ecosystem determine hydrological parameters such as rootzone storage, interception capacity,
infiltration capacity and subsurface drainage. Spatial variability of rootzone storage determines the
Beta function of the non-linear rootzone reservoir (Gao et al., 2018). This results in hydrological
response units based on landscape and geology (defining model structure), ecosystem and climate
(defining parameter values), which can be grouped per sub-basin.
Recession time scales can be derived from limited observations, if available, or otherwise
estimated; they do not affect the overall water balance. The longer time scales of groundwater
recession may be derived from Gravity Recovery and Climate Experiment (GRACE) data, which
can also be used to constrain groundwater dynamics (Winsemius et al., 2006; Hulsman et al.,
2021b).
Minor calibration parameters remain, such as the splitter between fast subsurface runoff and
recharge. These have a limited effect on the water balance and can be estimated if no observations
are available.

# 5   Why is the soil-based modelling tradition so rooted in hydrology?

## 5.1   Agricultural bias

Since hydrology was born from chapters of agricultural and hydraulics textbooks (Rodríguez-
Iturbe and Rinaldo, 2004), the "agricultural bias" has probably played a major role in
overemphasizing the importance of soils. In agriculture, the focus is on seasonal crops. A seasonal
crop has limited time to develop a rootzone storage that can buffer for longer term variability. At
best, it can buffer for average dry spells that may occur within an average year. This is why
modern agriculture requires water management by the farmer to buffer for natural fluctuations. In
agriculture, ploughing destroys preferential infiltration and sub-surface drainage. It also limits the
rootzone storage capacity to the relatively small soil layer above the plough pan. In such cases it is
indeed the moisture holding capacity of the soil that determines the rootzone storage capacity.
The widely used Penman-Monteith equation for estimating reference evaporation works well in
agriculture, where the dominant evaporation is from crops. However, it is likely not appropriate to
describe land-atmosphere interaction of natural ecosystems. Unfortunately, this "agricultural bias"
only applicable in small proportion of terrestrial area has been dominant in most hydrological
work. We argue that this deeply rooted soil-based perception may limit or even mislead the further
development of hydrological science, especially for next generation professionals.
Even in the Anthropocene, where human impacts on essential planetary processes have become
profound, and hydrological processes are affected by human activities such as agriculture,
urbanization and deforestation, we believe it is still essential to emphasize the importance of
ecosystem understanding. There are two reasons: 1) the majority of our earth, and particularly the
uphill runoff generating parts of catchments, is still dominated by natural ecosystems, although
human modification has modified 14.5% or 18.5 M $km^2$ of land (Theobald et al., 2020), and 2)
also for human modified systems the ecological approach applies, provided that the ecosystem is
given sufficient time to become self-sufficient and manage its own resources.
## 5.2   Unreliable intuition
Hydrologists intuitively see the soil as the critical agent. It may very well by the perspective of the
ant that causes it. As people, we are biased by our perspective and the scale at which we observe
processes. We are therefore just too small to perceive the larger scale processes that dominate
landscape hydrology. We tend to dig holes in the Earth and try to infer larger scale behaviour from
what we observe inside this hole. The human scale prevents us from seeing the larger picture. We
need the giant's perspective to recognise the patterns present in the landscape.
At the human scale, assuming that soil properties, such as texture and porosity, matter makes
intuitive sense. People tend to describe what they see, and if they see water flowing or
disappearing in the ground they think that it is because of such soil properties. The role of the
ecosystem as the driver of the system is much more difficult to recognize, especially within its
evolutionary history. It requires seeing the environment as a living organism, which continuously
evolves and adjusts to changing circumstances. It also implies that the hydrological properties are
not constant over time. The rootzone storage, the most critical control on rainfall-runoff processes,
is continuously changing in response to changing climatic and human drivers (Nijzink et al., 2016;
Bouaziz et al., 2022). Instead of describing the 'now' as an invariant and static condition, with
environmental properties as a given, we have to think of the history that determined these
environmental conditions, which is much more difficult to realise.

## 6   Limitations

We stress that our ecosystem approach is subject to certain limitations. First, it applies at the so-
called ecosystem scale. This spatial scale can vary depending on the environment. It can be a few
square meters for grass, in the order or hectares for forest, even larger for sparse vegetation.
Catchment scales are usually larger than the ecosystem scale. Therefore, our approach is generally
suited for scales that are typical in hydrological modelling application. Second, we are talking
about ecosystems that have reached a certain level of equilibrium and are self-sustained. We are
not limiting ourselves to natural ecosystem. They can also be artificially induced. But they do not
need to rely on artificial help for their survival, such as irrigation or fertilization. Third, our
arguments are mostly related to water fluxes, and they do not pertain to water chemistry. The
variability of soils can have a pronounced influence on predicting water quality, solute transport,
and transit times (Weiler et al., 2017; Sternagel et al., 2021).

## 7   Conclusions

Traditional hydrological theories place soil physical properties at the heart of hydrology,
considering them as the driver of water fluxes, which is misleading for both process understanding
and model development. In contrast we need an ecosystem-based approach, where the structure of
the terrestrial ecosystem and its internal processes are seen as a consequence of ecosystem water
management strategies needed for its survival and growth. Hence, the ecosystem is the ultimate
manager of the soil. We advocate a change in perspective that places the ecosystem and landscape
at the heart of terrestrial hydrology and develop holistic and alive ecosystem-based hydrological
models with a more realistic representation of hydrological processes.

**Acknowledgement**: This research has been supported by the National Natural Science
Foundation of China (grant nos. 42122002, 42071081). We are grateful to Adriaan J. (Ryan)
Teuling, Luca Brocca, Conrad Jackisch, and one anonymous referee, for their constructive and
detailed comments and suggestions, which have been greatly helpful for sharpening the message
and improving the flow of arguments.
**Code/Data availability:** No code or data sets were used in this article.
**Author contribution:** Three authors contributed equally to this work, including
conceptualization, writing, and revision.
**Competing interests:** At least one of the (co-)authors is a member of the editorial board of
Hydrology and Earth System Sciences. The peer-review process was guided by an independent
editor, and the authors have also no other competing interests to declare.

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
