# Peer review of "HESS Opinions: Are soils overrated in hydrology?"

_EGUsphere, 2023_

## Referee Comment (RC2)

[referee-annotated manuscript omitted]

---

## Author Comment (AC3)

Hongkai Gao and co-workers present an interesting contribution to the debate about key elements in the concepts of hydrological modelling. As an opinion paper, the authors argue that the affinity of hydrological model concepts to soil properties are more a relict than a substantial information basis. They propose to shift focus to the rootzone (as manifestation of the ecosystem) as alternative conceptual foundation.

**RC1.1:** I congratulate the authors for their work and I agree that our community has to keep challenging the conceptual assumptions and traditions. The role of soils in hydrology and land surface modelling is a particularly interesting debate. Recently Novick et al. (2022) have pointed to a "water potential information gap" in similar notion but opposite proposals. Discussing the role of pedotransfer functions (Looy et al. 2017, Vereecken et al. 2022) and soil hydraulic functions (Peters et al. 2023) together with structural adequacy of models (Gupta et al. 2012), perceptual model consistency (Wagener et al. 2021) and the data flow in model building (Gharari et al. 2021) and model analyses (Loritz et al. 2018) is in my view very important and promising. Hence I see the topic of this manuscript as worth an opinion paper.

We thank Conrad Jackisch for endorsing the need of our work, and his pointing out the relevance of this topic in relation to various developments in hydrological science and land surface modelling.

**RC1.2:** However, I am not really convinced that the current arrangement of the arguments in the manuscript is really substantiating this timely debate. My main concern is that the authors use the word "soil" for different concepts and at different scales without much differentiation. The critical zone concept (Lin et al. 2006) was already much further than this. Also the debate about landscape organisation and hydrologic functioning (Jackisch et al. 2021) including a critical assessment of conceptual assumptions about processes and scaling is more advanced on the topic.

We thank the reviewer for referring to previous work that is relevant to our discussion, and we will include these references in our revised version. However, we generally disagree with statements such that Lin et al. 2006 'goes much further than this' and that Jackisch et al. 2021 'is more advanced on the topic', which imply that our opinion paper is contained in these previous discussions.

The cited papers argue in favor of interdisciplinarity, recognizing that there is an intersection between hydrology and pedology (Lin et al. 2006) or landscape organization and hydrological functioning (Jackisch et al. 2021). We shall acknowledge and cite the relevant papers in our MS, such as on hydropedology (Lin et al. 2006), landscape organization and hydrology (Troch et al., 2015). We fully agree that the cross-fertilization of related disciplines can lead to novel knowledge, and that it is important to encourage a broader earth system science perspective to better understand interactions and feedbacks between different processes. We don't claim that we are the only ones, or that we invented the ecosystem centered approach. There is extensive previous work on these subjects, going from entire special issues in HESS dedicated to catchment co-evolution (https://hess.copernicus.org/articles/special_issue207.html) to the Gaia hypothesis. It's encouraging that there are more people who share this view. It is just that mainstream (often global scale) modellers continue to use the traditional soil-based approach (e.g. the recent paper by Vereecken et al. 2022).

However, our commentary should not be mistaken for an invite to look at things more broadly. It addresses some of the limitations in our ability to integrate different disciplines, by trying to disentangle some specific dependencies that are not fully understood. In particular, we are identify a dependency structure between climate, vegetation and soil hydraulic properties, which clarifies what one should

model and for which purpose. Therefore, compared to this previous work, our paper is more specific, but also more purposeful.

The picture below could perhaps help clarify our contribution. The overlapping circles shown on the left reflect the traditional view, which recognizes that climate, vegetation and soil all play in important role in governing catchment hydrological processes and cross influence each other. The nested circles shown on the right are closer to our view, where climate sets the boundaries for vegetation, and in turn, vegetation manages the soil hydraulic properties. We claim that for catchment hydrology it is unnecessary to dig into what happens in the 'soil circle', as the 'vegetation circle' provides the necessary level of detail.

[Figure]

Then there is the question of scale, which is often brought up when debating appropriate process descriptions. The paper of Vereecken et al., 2022 (e.g. in Figure 1) clearly indicates that one needs to understand soil processes at the pore scale in order to understand processes at larger scales (pedological, regional, global). This is a very common view, which we strongly oppose, as it fails to recognize that a natural system exhibits emergent properties, which effectively enable a description of large-scale processes independent on what happens at the smaller scale. We shall clarify that our main focus is describing hydrological behaviour at system scale, ranging from the watershed scale to the catchment scale, hence considering topographic areas that are large enough for a stream channel to be identified, which typically takes at least a few hectares. This scale is clearly too large to enable a detailed characterization of pore scale processes, but fortunately sufficiently large so that considerable process integration takes place, which makes it possible to characterize the overall system behaviour through its emergent properties.

**RC1.3:** The authors do address such aspects in their manuscript and point to intertwined factors and sub-systems. However, the arguments are not really brought to consistently support the very fundamental claim of the manuscript. Without meaning to offend the authors, I would see many of the claims rather being rooted in conceptual limitations in the view of soil functions by the authors than in

the lack of information or importance of soils in hydrological processes and models. I will substantiate this in the more detailed assessment.

We do not agree that our abandoning detailed description of soil processes is caused by a lack of understanding of such processes. The point is that detailed knowledge does not prevent us from seeing the larger scale picture. We reply to the individual points in the more detailed comments.

**RC1.4:** In general, I do not really see, how the replacement of a soil-centred with a rootzone-centred concept deviated from the critical zone concepts (Lin et al. 2006). I also do not see, why the authors omit the main driving concept for fluxes (depletion of gradients) and thus the whole debate about potentials (Novick et al. 2022). I would have liked to see to which degree their arguments are essentially an expression of the conceptualisation of hydrological models i) as distributed and linked storages, ii) at a broader scale (in the sense of the scale triplet) and iii) with soils expressed by texture classes.

The paper of Lin et al. 2006 introduces hydropedology as an integration of pedology and hydrology. The first difference between Lin's and our work is in the system of interest. Their system is the root and deep vadose zone, where the relevance for hydrology is the description of flow pathways in the unsaturated zone. Vegetation is only mentioned tangentially in their work. Our interest is in the hydrological system, and in particular, the relationship between precipitation and streamflow. Vegetation is an integral part of this system and the ultimate water manager, as an active agent. As a result, we are interested in understanding the overall behaviour of the root zone. They are interested in what happens inside it.

The second difference between Lin's and our work is that pedology and hydrology in Lin's work are seen at the same level, in the sense that they can cross-influence each other, which is true at the scale they are primarily interested in. In our work, we establish a hierarchy, where it is the hydrology – by means of the ecosystem as the active agent – that influences the soil much more than vice-versa. Again, this applies to the overall behaviour of the root zone at the catchment scale.

Novick et al. 2022 is about potentials that govern the water flows throughout the soil–plant–atmosphere. These processes are much more specific than the level of detail we are focusing on in our commentary which is at larger scale hydrological behaviour.

In order to avoid these misunderstandings, we will clarify in our revised version that we are primarily interested in the description of hydrological processes at system scale.

**RC1.5:** Moreover, I find many claims very strong and confrontative (e.g. L21f, L111) and not well-balanced. To really spark the debate (and not a battle) I would have liked a more balanced and substantiated formulation.

Our paper is meant to trigger debate and definitely not a battle. However, we also feel that an opinion paper should include clear views, personal thoughts, and opinions that are not universally shared. Such opinions may be considered provocative and may not please everyone, but if the arguments are too balanced, then there is not much incentive for debate.

Detailed comments:

**RC1.6:** L43: I would argue that this is the debate throughout in pedology. At least not only recently.

"Soil forms an ecosystem in itself" is a common sense in pedology. This can be found in many soil textbooks, such as "The Nature and Properties of Soils. 15th edition, Weil and Brady, 2017". As suggested by the reviewer, we will remove "recently".

**RC1.7:** L49: Why do you limit the perspective to abiotic boundary conditions when you actually argure for an ecosystem perspective? Biodiversity, niches, disturbances, stressors, carbon pools are not all determined by climate and geology. Moreover, at least most temperate soils do not develop directly from bedrock material but on deposited material from rather old geomorphological processes (which include path-dependent development options). Pointing to this, soil degradation and soil loss too is an important and largely irreversible process with severe implications on regional hydrological and biogeochemical cycles.

We shall clarify that we are referring to systems that are vegetated, and where the ecosystem had the time to adapt and is in a condition of a relatively stable equilibrium. Such systems, we would argue, have the ability to adapt to 'disturbances', such as heavy rains that can cause soil erosion, or dry periods that can cause water shortage. The reviewer is right to mention that there can be disturbances that can be destructive and can determine irreversible processes. These extreme cases are not the focus of our commentary. However, we believe that a better understanding of the behaviour of an ecosystem in its 'mature' state can also help understand when tipping points are reached that can cause such irreversible processes.

**RC1.8:** L59: I think I have an idea what you intend to express, but since this simplistic/reductionist pedotransfer approach has a couple of implications which could be challenged. I suggest to clarify this sentence a little more and link to the debates in soil physics and the pedotransfer community.

We will rephrase. Our intention was not to criticize pedology as a discipline, but rather, the way that concepts developed within soil science are utilized in hydrology.

**RC1.9:** L63 (opposite): I am not sure if I can follow. The argument before was that simple pedotransfer and soil hydraulic property models are an issue and that they become coupled to rooting depth.

Consequently, rooting depth has to be defined somehow to eventually assess plant-available soil water storage. Why does it matter if in this step plant-available soil water storage becomes the dependent variable of the other involved variables, if it is used as independent variable in the proceeding calculation steps? Isn't this a question about the perceptual model underlying any form of conceptualisation and numerical expression?

L63 (root zone storage): Yes, but maybe at different time scales? Plants and ecosystem may adapt and co-evolve (within a range of their survival). So why should the debate be solved by exchanging the depending/independent variables? Could this actually be a scaling issue?

This is a good question. Whether regarding root zone water storage as a dependent or independent variable, results in very different perceptual hydrological models. Traditional models regard root zone water storage as a function of soil moisture and rooting depth, which is a typical reductionist perspective. Such an approach requires to collect endless spatial and temporal heterogeneities of soil, water, and underground biomass variations to obtain root zone water storage. There are two limitations

of this method. 1) Data availability. These small-scale details are usually not available especially at large scale. This limitation "will remain unresolved in the foreseeable future" (Or, 2019). 2) more importantly, the ecosystem is not a stationary item. Ecosystems can and do adapt to changes. Hence, rooting depth is a dynamic variable in response to climatic and anthropogenic drivers.

It is not the combination of the moisture holding capacity of the soil and the rooting depth that determines how much water plants use, but rather the opposite: the ecosystem has a certain water demand and thus requires a certain root zone water storage as a buffer for critical periods of drought. This storage can be reflected in root density, depth and lateral extension, but it boils down to a hydrological volume that can be easily conceptualized in a hydrological model.

Regarding the scaling issue, we refer to our answer to comment RC1.2.

**RC1.10:** L65: This might depend on what exactly we see as detailed. As you open your argumentation with coevolution, maybe a broad idea about the general type of soil (not texture class) and biome (including its ecohydrological properties) could be sufficiently detailed? If so, remote sensing claims various solutions to gather such data…

We agree that some soil properties such as soil type could in principle be mapped, and there are soil databases that offer such information in some parts of the globe. However, some of the variables that hydrologists need are very difficult to obtain, even with detailed mapping. Concerning vegetation modelling, one of the key variables is rooting depth, which together with wilting point and field capacity, would determine plant available storage. We will further clarify these statements in the revised version.

**RC1.11:** L67: Again, i would see this as a scale issue: Ecosystem and climate are both terms referring to large scales (in the sense of the scale triplet). Hydrology is not referring to a specific scale.

We agree that we are referring to a larger scale. The scale triplet referred to by the reviewer is the 'Process scale', 'observation scale' and 'modelling (working) scale' defined by Bloeschl and Sivapalan (1995). As clarified earlier, we are considering processes at the system scale. This is also our modelling scale, as we are interested in characterizing the processes directly at this scale. Unfortunately, there is a well-known lack of direct observations at this scale. We will clarify the scale issue in our revised version.

**RC1.12:** L72f: This assumes that the ecosystem is somewhat in equilibrium with determinable drivers of its development. However, path-dependent trajectories, dynamic deviations from equilibrium more or less buffered by the ecosystem and any application for global changes (climate, land use, cohabitation…) but severe challenges to this view.

It is correct that the method applies under the condition that the ecosystem has converged to some sort of equilibrium. Indeed, it does not apply to agricultural fields, as we specify later in the paper. Regarding predicting the time to equilibrium, this is a difficult issue, as it depends on how fast the change is and how rapidly the ecosystem can adapt to it. Apart from simulating observed spatial variability (Gao et al., 2014, de Boer-Euser et al, 2019), the method has been applied to simulate temporal variability in response to vegetation change (Nijzink et al., 2016) and climate change (Bouaziz et al., 2022). In these case study, it was shown that the method could be used to anticipate how the system adapts to such

sources of variability. The case of agricultural fields is an example where the system is not given the time to adapt and where the proposed method does not apply.

Nijzink, R., Hutton, C., Pechlivanidis, I., Capell, R., Arheimer, B., Freer, J., Han, D., Wagener, T., McGuire, K., Savenije, H., and Hrachowitz, M.: The evolution of root-zone moisture capacities after deforestation: a step towards hydrological predictions under change?, Hydrol. Earth Syst. Sci., 20, 4775–4799, https://doi.org/10.5194/hess-20-4775-2016, 2016.

de Boer-Euser, T., Meriö, L.-J., and Marttila, H.: Understanding variability in root zone storage capacity in boreal regions, Hydrol. Earth Syst. Sci., 23, 125–138, https://doi.org/10.5194/hess-23-125-2019, 2019.

Bouaziz, L. J. E., Aalbers, E. E., Weerts, A. H., Hegnauer, M., Buiteveld, H., Lammersen, R., Stam, J., Sprokkereef, E., Savenije, H. H. G., and Hrachowitz, M.: Ecosystem adaptation to climate change: the sensitivity of hydrological predictions to time-dynamic model parameters, Hydrol. Earth Syst. Sci., 26, 1295–1318, https://doi.org/10.5194/hess-26-1295-2022, 2022.

Gao, H., M. Hrachowitz, F. Fenicia, S. Gharari, and H. H. G. Savenije, 2014. Testing the realism of a topography driven model (FLEX-Topo) in the nested catchments of the Upper Heihe, China, Hydrol. Earth Syst. Sci., 18, 1895-1915, 2014.

**RC1.13:** L78: Why do you refer exactly to these citations? I would think that e.g. the work of Gardner including his famous lab experiments have been far more important for propagating this perception.

We will consider more references in the revised paper.

**RC1.14:** L87ff: Yes, and this might be one of the actual issues to address here. Linear Darcy filter flow has been coupled to highly non-linear retention properties with the Richards equation and as a first-order diffusive flow model, it does an ok job for diffusive flow in somewhat well-defined porous media. However, especially infiltration (as initial soil water redistribution into the soil during rain events) is often not dominated by diffusive flow but by advection (Newtion Shear Flow equation (Germann, 2020), soil moisture velocity equation (Ogden et al. 2017), particle model (Jackisch and Zehe 2018), non-equilibrium flow (Vogel et al. 2023)). To my understanding, this deficit is rooting back to the very limited means to measure antecedent state-dependent infiltration and to use such data in hydrologic models. But why this is an argument for soils not being central in the question for one of their fundamental services to mediate the local soil water cycle is not clear to me. Especially because infiltration is state dependent, precipitation may not be retained after drought conditions, requiring vast amounts of light rains, slow snow melts or similar to replenish the water stocks, while storm events will simply lead to preferential flow and possibly erosion…

We agree that soil fulfils several important functions. Provided that we are considering catchment scale processes, and therefore integrating processes that take a minimum area to be operative, the question is whether processes such as infiltration, retention or release to subsurface flow depend on soil properties such as texture or can be somehow related to them, as many hydrological models assume. Our suggestion is that the processes that are significant at the catchment scale are conditioned on a multitude of soil properties, such as macropores, rootzone depth, etc, which are themselves conditioned on the vegetation, which ultimately is conditioned by climate, as shown in diagram above.

In order to model the catchment scale processes, in this set of nested dependencies, it is sufficient to stop at the vegetation level. There are other applications, however, where it is necessary to dig into the soil level.

**RC1.15:** L93: Well, it is not dominant when it comes to storm events, yes. But these experiments use rather steep gradients with a lot of water. The debate about when and to what degree soil water flow is preferential is ongoing. If this was the full story, soils could hardly sustain the ecosystems.

We shall clarify that we do not negate the existence of matrix flow, which is how the soil becomes wet. However, preferential flow is observed ubiquitously, and is critical to sustain ecosystems, as this is the way ecosystems get rid of excess water, which would otherwise favour soil erosion or create waterlogged conditions where roots would deteriorate. For example, in a Mediterranean climate, isotope tracing has shown that at the beginning of the wet season, preferential flow allows precipitation to rapidly replenish ecosystem water deficit. While during wet periods, the excess water after precipitation is drained efficiently by preferential flow and subsurface drainage (Brooks et al., 2009). By the way, this is also a perfect example of "catchment as a living organism".

Brooks, J.R., Barnard, H. R., Coulombe, R. and McDonnell, J.J. Ecohydrologic separation of water between trees and streams in a Mediterranean climate. Nature Geosciences, 2009

**RC1.16:** L98: Partly yes. But preferential flow can also simplify our models. Anyways, I suggest to ease the dispute opened by this statement with a slightly more balanced view on achievements towards unifying forms of non-uniform infiltration.

We will consider your suggestion here to make our statement more balanced.

**RC1.17:** L108: Yes. But this again can be seen as a scaling issue. At the hillslope- and plot-scale, these parameters/concepts have been very central. Only at the catchment-scale they could be easily subsumed as general soil property parameters not requiring for a dual domain definition. And this is true for the hindcast of our observations…

We will clarify that we refer to the catchment scale as mentioned in the comments above.

**RC1.18:** L111: Again a strong claim. I can agree that the pref flow debate has always struggled to connect to Darcy-scale soil physics. But fog? No progress? Is this claim really needed for your argument?

The progress in preferential flow studies are discussed in L92-108. We will also add what the reviewer recommends about the importance of advection flow. To our best knowledge, the importance of preferential flow has been recognized at least to Schumacher (1864) (Beven 2018). Still until now, we did not see clear improvement of preferential flow models in runoff prediction practice. Hence, we chose on purpose a strong claim.

**RC1.19:** L113 (a priori assumption): Well, they are ABOUT the description of soil water flow. If they are key for describing hydrological processes is part of the actual model conceptualisation, its numerics and the respective regimes under study. Again, I would argue that this is no other "a priori assumption" as most other parts of the perceptual model. And since its actual effect in the model can be and is challenged (Glaser et al. 2018), I would rather see it as a positive example for advancing hydrologic models.

Yes, we agree that all our perceptual models need experiments to test and sharpen. Both accepting and rejecting a-priori assumptions is good for advancing our understanding of the hydrological system.

**RC1.20:** L118: So far soil variability has not been motivated. This is especially difficult, because the effect of soil variability is again a matter of scale (including the respective range of processes). After reading subsection 3.1, I can think of quite a number of papers, providing good evidence for the opposite: When you have the average soil right, you can easily reproduce observed hydrologic patterns (e.g. Loritz et al. 2017).

As mentioned above, we will clarify the scale issue.

**RC1.21:** L123f: Ok. Known and well established. Maybe citing some of the many studies would be nice.

These results are not from other literatures, but they are calculated from the state-of-the-art ERA-5 reanalysis data. There are similar studies, which we will cite in the revised MS.

**RC1.22:** L131f: This argument is not really sound. Studies fully agree that plants and ecosystems strongly moderate the net ET flux of a stand. But without soil as the part of the ecosystem which can actually store water for weeks and beyond, this percentage cannot be reached. We exactly see this in data based on Budyko-like assessments that more draining locations (sandy, karstic) have very little ETact simply because precipitation is largely drained.

We agree that the soil supports the ecosystem, and we stated this from the outset. However, we also believe that vegetation will adapt to the soi conditions. For example, special geological condition in Karst region, i.e. the carbonate rock, determines very thin soil layer. This is not an exception, but another good example, that soil is the consequence of co-evolution of ecosystem in certain climate and geology. Also our proposed ecosystem-centred rootzone method has been used in Karst and other mountainous regions. The reviewer can refer two recent studies (Gao, 2020; McCormick et al., 2021).

Gao, Y. Modeling Flow and Nitrate Transport in Karst Groundwater Basins. University of Central Florida, PhD thesis, 2020

McCormick, E. L., Dralle, D. N., Hahm, W. J., Tune, A. K., Schmidt, L. M., Chadwick, K. D., and Rempe, D. M.: Widespread woody plant use of water stored in bedrock, Nature, 597, 225–229, https://doi.org/10.1038/s41586-021-03761-3, 2021.

**RC1.23:** L137f: Yes, difficult but steeply advancing. Please see Peters et al. (2023) and Hohenbrink et al. (submitted to ESSD) for examples. The most critical part might be the reduction of such data to van Genuchten/Mualem SHP model parameters and the weakly informative relation to the broad texture classes, BD and Corg. But the issue of pedotransfer models is a discussion on its own, and which is currently gaining momentum.

We will cite more recent papers.

**RC1.24:** L144: The issue here might be that soil mapping is not particularly done for hydrological purposes. On the one hand, pedological classes are not always directly convertible to hydrological properties. On the other hand, soil stratification and the respective hydrological properties are rarely conveyed into land surface models with sufficient degree of vertical resolution. Moreover, the uncertainty about the hydrological properties of the mapped soil classes is largely unknown and very

different from region to region. Given all of these points, I am not quite sure if "interpolation and upscaling" is the core issue here. Maybe it is more a disconnection between soil mappers and hydrologic modellers?

We agree that soil maps are not particularly useful for hydrology. There have been attempts to develop hydrological soil maps such as 'the hydrology of soil types' (Boorman et al, 1995) in UK, but these are not widely available. However, our main argument is that pursuing that route is unnecessary if one is interested in describing catchment scale processes. This does not mean that a connection between soil mappers and hydrological modelers would not be fruitful, particularly if the focus is on describing smaller scale processes.

**RC1.25:** L148: I fully agree that unnatural lab conditions are a fundamental difficulty. However, many measurements are conducted using "undisturbed" samples for soil hydraulic property analyses. It is unnatural because the samples are extracted from their capillary context, exposed to free evaporation at the surface and a no flow boundary at the bottom (for the standard HYPROP protocol). However pedotransfer functions are then correlating lab measurements (soil hydraulic properties) to lab measurements (texture) and the scaling and transfer involved in its application to field conditions remain hidden.

We agree that with a lot of effort one can obtain reliable pedotransfer functions, however, this is impractical at the catchment scale. Moreover, such functions are typically surface maps, and lack a vertical dimension, which is necessary to model what happens in the underground. The pedotransfer functions approach is plagued with uncertainties and difficulties of various kinds. For this reason, we believe that exploring alternative, potentially easier approaches is worthwhile.

**RC1.26:** L152: I fully agree, but again this is an issue with quite a bit of literature from hydropedology to cite here.

Will search and read.

**RC1.27:** L153ff: I do not get this point. The discussion about parameter regionalisation has a long standing in hydrology. E.g. mHM (Samaniego et al. 2017) exactly works because it modifies the initial lab scale parameters to match its distributed effects on fluxes in the landscape. Showing one odd model result can have so many reasons that I find it very difficult to support your argument through it.

The Netherlands likely benefits from the most detailed soil survey in the world, thanks to its very advanced agricultural science and technology. But even with such detailed soil data, the soil-based evaporation model produced such a large discrepancy. This is a strong indication that collecting detailed soil data does not benefit hydrological studies. This is not one odd model result, but a strong example showing the dead-end track of this methodology.

**RC1.28:** L157: Which is a nice example for model extrapolation and the shift in parameter sensitivity under climate change (Melsen and Guse 2021).

Thank you for sharing this literature. We will read it.

**RC1.29:** L177: Again, a difficult claim. They test if texture classes and soil depth is informative. However Novick et al. (2022) point nicely to soil water potential being most informative and often omitted in LSMs. The model you are referring to are not particularly strong in soil physics as they conceptualise soils as stores instead of any framework of potentials as drivers. So your assessment might actually pinpoint that soil hydrology based on a storage concept is not very informative? As stated in the general section, I find it very difficult that you do not discern between weak conceptualisations of soils and the actual physical properties and dynamics linked to soils.

Please refer to our reply to RC1.4.

**RC1.30:** L188: These intertwined factors mostly manifest at "soil scales", which are not necessarily very small.

Our attitude to terminology in this opinion paper can be found in RC1.13.

**RC1.31:** L194f: Again, I would argue that the concept of infiltration capacity as rigid site property maybe the root of the issue here? Infiltration capacity to my understanding does not necessarily entail a constant or any specific model (e.g. Horton which is subsuming site properties and antecedent condition into an exponential decay function for infiltration rate or Green and Ampt which indeed is rarely proven in natural soils). Since infiltration is the passage of water into the soil domain, I would argue that soil structures (draining macropores and storing finer pores) facilitate it and that antecedent conditions plus the rainfall supply dynamics govern the individual initial (non-uniform) soil water redistribution (see comment to L87ff). The ecosystem modifies the boundary conditions, state dynamics and structure formation in the long run (Lange et al. 2015 and other publications from the Jena experiment).

The reviewer's thought on infiltration capacity is interesting. We agree that "infiltration capacity… does not necessarily entail a constant or any specific model", which is in line with our augment to question the long-term held belief that soil determines infiltration capacity. The concept of infiltration capacity is still important in hydrology, especially for storm events. But in natural hillslope and catchments, vegetation, topography and other land surface are all indispensable factors in storm event modelling and may play a more significant role controlling the infiltration capacity than soil properties.

**RC1.32:** L203ff: I agree and I admire the authors for their very nice contributions to these examples. However, this comparison is not fair since the intended applications of more complex models are often more than rainfall-runoff modelling. Especially when models are used to analyse effects of changes in land use , climate regime, management etc. the stationarity assumption collapses and we require parameters and submodels with physical meaning. Once we have a good understanding about how the modified hydrologic system can be conceptualised, the simple models are much more efficient and maybe even less error-prone again. But the transition (in system characteristics or scale) remains very challenging for these kind of models.

The reviewer might not get our points. We believe the reviewer is also with us. Both of us agree that we should not develop our model based on stationary assumption. But the soil-based model is a typical stationary model, since soil properties are mostly stable and unchanged with climate and human activities in short term. What changes dramatically are the land use and land cover and belowground biomass in the background of both human activities and climate change. Our proposed ecosystem-based model can deal with this issue much better than soil-based models. Because ecosystem-based model

intrinsically regards catchment as a living organism. The reviewer may refer to many published papers (Nijzink et al., 2016; de Boer-Euser et al., 2019; Bouaziz et al., 2022).

**RC1.33:** L210f and Fig. 2: I do not find it a logical proof of your argument that some models can succeed without soil information. If soil information is only texture class and porosity maybe it is more telling that these properties are not very informative for hydropedological characteristics and that the variable for the most frequent antecedent conditions (aridity) has far more influence because it is more informative for hydrological functioning? Hohenbrink et al. (submitted to ESSD, soon at https://doi.org/10.5194/essd-2023-74) show very nicely how these standard properties and texture-based soil classes do not inform hydropedologic functioning.

Soil texture is the most easily accessible soil information, that could be the reason Addor et al (2018) chose these characteristics to compare with hydrological signatures.

**RC1.34:** L217ff: I find it difficult to discern your "ecosystem"/"rootzone" approach from the hydropedology concepts (Lin et al. 2006).

Root zone and critical zone have strong connections, but with obvious differences. The lower and upper boundary of critical zone is still debated. But usually, Earth's critical zone includes air, soil, water, rock and organisms. For hydrology, the ecosystem with its rootzone is the most active layer in the critical zone. For example, in the Loess Plateau where soil is thick, root zone is merely the active layer on the topsoil. In Karst and other mountainous regions, rootzone includes not only the soil water storage, but also the fissure water storage in bedrock. In very dry climates, roots can even reach the deep groundwater, thus in this case, the rootzone also includes some part of the groundwater (see Singh et al., 2020). In cropland, where irrigation provides an extra water supply to rootzone during dry seasons, the rootzone water storage capacity is often smaller than under natural conditions with similar climate background. The rootzone is the most active layer in the critical zone (with as much or even more biomass than above ground) controlling land surface processes, including hydrology.

Singh, C., Wang-Erlandsson, L., Fetzer, I., Rockström, J., and Van der Ent, R., 2020. Rootzone storage capacity reveals drought coping strategies along rainforest-savanna transitions, Environ. Res. Lett. 15 (2020) 124021

**RC1.35:** L226ff: Within the lines of arguments, I think you are jumping through different scales here (with concepts and properties which are known NOT to be scale-invariant). The assumption that the ecosystem will be able to become the dominant driver is only true if the system has sufficient degrees of freedom to do so. Mediterranean basins have been deforested long ago, soil has been lost and there is no sign of spontaneous ecosystem replenishing under the current climate conditions. Badlands, crusts, long-term unstable debris are examples contradicting your claim. Hence a more differentiated analysis would be more insightful?

We will clarify the scale issue, as mentioned in the comments above.

**RC1.36:** L232: I fully agree that water can bypass the rootzone but is not necessarily reaching groundwater. In many soil systems of the mid latitudes we find laterally conductive layers formed by more distant ice ages leading to relatively quick drainage or even interflow. Your FLEX approach has nicely shown this for the Ardennes…

We will add subsurface storm flow in the revised MS.

**RC1.37:** L235ff: With having FLEX in mind I can understand your reasoning but I find your PERCEPTUAL model rather inflexible in the first place. The notion to simplify as much as possible is fully legit but deterministic concepts are in my understanding rather a thing of the past when we were limited in computational powers. And I find that this stiffness weakens your argumentation.

We use FLEX merely as an example, but our approach is not constrained by the modelling concept and is not an advertisement for simple models. The reviewer misunderstood the intention of the paper. We state that the ecosystem should be central in all hydrological modelling at whichever scale, since the ecosystem is the active agent that reacts to and modifies the impact of external drivers. A nice consequence of this fact is that models can often be much simpler than the modeler intuitively assumes, but it is not a requirement.

**RC1.38:** L245 (and the paragraphs before and after): I do not see why this is an argument against the importance of soils. Just because modellers use non-informative variables about soils and just because they have not found laws to scale the scale-dependent concepts/models does not mean that soils are not important. If these observations are biased, this does actually point to a misconception of the soil system rather than serving as an argument for omitting soils altogether. I would claim that this only shows that soil function cannot be described by texture classes (alone).

It is worthwhile to note again that we did not intend to omit soils altogether. We claimed that "Soil is important" at the very beginning of our paper. We proposed to considering root zone as an integrated system, rather than simply treating soil and roots as isolated parts.

**RC1.39:** L251: I find it very difficult to agree to your arguments at this too general level of characterisation of somewhat arbitrarily selected model examples. I suggest to build the arguments based on the state of the art about structural adequacy and model conceptualisation (see general comments)

Please see our response to your general comments.

**RC1.40:** L282ff (and the whole subsection): You are proposing a new conceptualisation in which you omit various central properties governing water retention and drainage, which are not only governed by vegetation alone. With most of the terrestrial surface of our planet being actively managed by humans and a massively changing climate and biosphere, I find it not very helpful from a physical and system perspective. Moreover, your concept does not evade the scale issues. Quite to the contrary the active rootzone is not a static thing (at many scales). When we look at root water uptake alone, the sourcing depth of water within the root zone is dynamic over the year and very different from site to site (with the very same tree species and ages) (Jackisch et al. 2020). Giving reference to ERA5 data for this is maybe a little too large scale to substantiate your arguments with?

We agree that active rootzone is not a static thing (at many scales), thus we need to develop an alive model to take these changes into account. This is exactly what we are saying in our opinion paper. Please refer to some of more references (Nijzink et al., 2016; de Boer-Euser et al., 2019; Bouaziz et al., 2022).

Our argument is mainly for catchment scales. The concept of preferential flow was proposed in small scale soil profiles, but hydrologists found preferential flow is everywhere for all hydrological processes at multi-scales (Uhlenbrook, 2006). Also the root zone is not only important for catchment hydrology, but also for land surface processes, and is essential for ecosystem's resilience to drought at multiple scales, including landscape, regional and global scale.

Nijzink, R., Hutton, C., Pechlivanidis, I., Capell, R., Arheimer, B., Freer, J., Han, D., Wagener, T., McGuire, K., Savenije, H., and Hrachowitz, M.: The evolution of root-zone moisture capacities after deforestation: a step towards hydrological predictions under change?, Hydrol. Earth Syst. Sci., 20, 4775–4799, https://doi.org/10.5194/hess-20-4775-2016, 2016.

de Boer-Euser, T., Meriö, L.-J., and Marttila, H.: Understanding variability in root zone storage capacity in boreal regions, Hydrol. Earth Syst. Sci., 23, 125–138, https://doi.org/10.5194/hess-23-125-2019, 2019.

Bouaziz, L. J. E., Aalbers, E. E., Weerts, A. H., Hegnauer, M., Buiteveld, H., Lammersen, R., Stam, J., Sprokkereef, E., Savenije, H. H. G., and Hrachowitz, M.: Ecosystem adaptation to climate change: the sensitivity of hydrological predictions to time-dynamic model parameters, Hydrol. Earth Syst. Sci., 26, 1295–1318, https://doi.org/10.5194/hess-26-1295-2022, 2022.

Uhlenbrook, S.: Catchment hydrology–A science in which all processes are preferential, Hydrol. Process., 20, 3581–3585, https://doi.org/10.1002/hyp.6564, 2006.

**RC1.41:** L295ff: From a (soil and hydrologic) physics perspective the main fundament might be that fluxes are driven by gradient depletion and that the degrees of freedom for these fluxes are state dependent (including subscale properties subsumed as hysteresis). The fill-and-spill concept (McDonnell et al. 2021) is a very powerful description of dynamic connectivity and threshold behaviour resulting from the strong non-linearities in soils. However, the depletion of gradients is largely omitted in such models. You might argue (L298f?) that storage-based models do not require an explicit treatment of gradients since it is all implicitly covered by the individual storage and transfer functions. However, this is not an argument against the importance of soils nor does it solve the standing issue to be capable to convey changing landscape properties into the required storage characteristics.

Please see our response to your general comment on gradient and fluxes.

**RC1.42:** L308ff: Why do you jump from the debate about the concepts back to the debate about available data (which has so far not been really opened)?

The landscape-based model we are proposing requires quite different data than soil-based models. What we are trying to say is that the data supporting landscape-based model development is booming. The increasing data accessibility allows us to test the model realism not only in terms of runoff, but also internal fluxes with more complementary data (Gao et al., 2020; Hulsman et al., 2021).

Gao, H., Dong, J., Chen, X., Cai, H., Liu, Z., Jin, Z., Mao, D., Yang, Z., Duan, Z. (2020). Stepwise modeling and the importance of internal variables validation to test model realism in a data scarce glacier basin. Journal of Hydrology. 591, 125457

Hulsman, P., Savenije, H. H. G., and Hrachowitz, M.: Learning from satellite observations: increased understanding of catchment processes through stepwise model improvement, Hydrol. Earth Syst. Sci., 25, 957–982, https://doi.org/10.5194/hess-25-957-2021, 2021.

**RC1.43:** L320ff: Since I read your manuscript as a strong claim for a simplified hydropedologic perceptual model, I find the argument with Occams razor very problematic. I would claim that we are in a situation with plenty of data to challenge our perceptual models and we have the tools to do this (e.g. Höge et al. 2020, Guthke 2017). Occams razor is a perceptual assumption, too.

We don't agree. Firstly, our model is NOT a simplified hydropedologic perceptual model. It regards root zone as an integrated system, rather than simply summarizing isolated parts together, e.g. soil, water, and roots etc. It is controlled by ecosystem's adaption to climate.

Secondly, Occams razor is not a perceptual assumption which can be tested, but a problem-solving principle in philosophy, which means "The simplest explanation is usually the best one." Although it is not a science hypothesis, it is widely used as a heuristic to guide scientific theory development. We believe this is very appropriate in an opinion paper to stimulate thoughts on model complexity or parsimony.

**RC1.44:** Again, I sincerely thank the authors for raising this debate. I hope that my review can contribute to sharpening the arguments and to raise awareness about the many aspects that might have fallen a little too short in preparing this manuscript.

We are very grateful to Conrad Jackisch for his very detailed and well-argued comments and for taking ample time to enter in this debate with us. We think and hope that our slightly provocative approach has stimulated the convergence of different viewpoints and hydrological schools.

We will incorporate all the valuable suggestions for improvement and will address omissions in the literature.

Bibliography

Germann, P.: Viscosity Controls Rapid Infiltration and Drainage, Not the Macropores, Water, 12, 337–15, https://doi.org/10.3390/w12020337, 2020.

Gharari, S., Gupta, H. V., Clark, M. P., Hrachowitz, M., Fenicia, F., Matgen, P., and Savenije, H. H. G.: Understanding the Information Content in the Hierarchy of Model Development Decisions: Learning From Data, Water Resour Res, 57, https://doi.org/10.1029/2020wr027948, 2021.

Glaser, B., Jackisch, C., Hopp, L., and Klaus, J.: How Meaningful are Plot-Scale Observations and Simulations of Preferential Flow for Catchment Models?, Vadose Zone J, 18, 0–18, https://doi.org/10.2136/vzj2018.08.0146, 2019.

Gupta, H., Clark, M. P., Vrugt, J. A., Abramowitz, G., and Ye, M.: Towards a Comprehensive Assessment of Model Structural Adequacy, Water Resour Res, 48, 1–40, https://doi.org/10.1029/2011wr011044, 2012.

Guthke, A.: Defensible Model Complexity: A Call for Data-Based and Goal-Oriented Model Choice, Groundwater, 55, 646–650, https://doi.org/10.1111/gwat.12554, 2017.

Höge, M., Guthke, A., and Nowak, W.: Bayesian Model Weighting: The Many Faces of Model Averaging, Water-sui, Water, 12, 309, https://doi.org/10.3390/w12020309, 2020.

Jackisch, C., Hassler, S. K., Hohenbrink, T. L., Blume, T., Laudon, H., McMillan, H., Saco, P., and Schaik, L. van: Preface: Linking landscape organisation and hydrological functioning: from hypotheses and observations to concepts, models and understanding, Hydrol Earth Syst Sc, 25, 5277–5285, https://doi.org/10.5194/hess-25-5277-2021, 2021.

Jackisch, C., Knoblauch, S., Blume, T., Zehe, E., and Hassler, S. K.: Estimates of tree root water uptake from soil moisture profile dynamics, Biogeosciences, 17, 5787–5808, https://doi.org/10.5194/bg-17-5787-2020, 2020.

Jackisch, C. and Zehe, E.: Ecohydrological particle model based on representative domains, Hydrol Earth Syst Sc, 22, 3639–3662, https://doi.org/10.5194/hess-22-3639-2018, 2018.

Lange, M., Eisenhauer, N., Sierra, C. A., Beßler, H., Engels, C., Griffiths, R. I., Mellado-Vázquez, P. G., Malik, A. A., Roy, J., Scheu, S., Steinbeiss, S., Thomson, B. C., Trumbore, S. E., and Gleixner, G.: Plant diversity increases soil microbial activity and soil carbon storage, Nature Communications, 6, 1–8, https://doi.org/10.1038/ncomms7707, 2015.

Lin, H., Bouma, J., Pachepsky, Y., Western, A., Thompson, J., Genuchten, R. van, Vogel, H.-J., and Lilly, A.: Hydropedology: Synergistic integration of pedology and hydrology, Water Resour Res, 42, 2509–13, https://doi.org/10.1029/2005wr004085, 2006.

Looy, K. V., Bouma, J., Herbst, M., Koestel, J., Minasny, B., Mishra, U., Montzka, C., Nemes, A., Pachepsky, Y. A., Padarian, J., Schaap, M. G., Tóth, B., Verhoef, A., Vanderborght, J., Ploeg, M. J., Weihermüller, L., Zacharias, S., Zhang, Y., and Vereecken, H.: Pedotransfer Functions in Earth System Science: Challenges and Perspectives, Rev Geophys, 55, 1199–1256, https://doi.org/10.1002/2017rg000581, 2017.

Loritz, R., Hassler, S. K., Jackisch, C., Allroggen, N., Schaik, L. van, and Wienhöfer, J.: Picturing and modeling catchments by representative hillslopes, 21, 1225–1249, https://doi.org/10.5194/hess-21-1225-2017, 2017.

Loritz, R., Gupta, H., Jackisch, C., Westhoff, M., Kleidon, A., Ehret, U., and Zehe, E.: On the dynamic nature of hydrological similarity, HESS 22, 3663–3684, https://doi.org/10.5194/hess-22-3663-2018, 2018.

McDonnell, J. J., Spence, C., Karran, D. J., Meerveld, H. J. (Ilja) van, and Harman, C. J.: Fill-and-Spill: A Process Description of Runoff Generation at the Scale of the Beholder, Water Resour Res, 57, https://doi.org/10.1029/2020wr027514, 2021.

Melsen, L. A. and Guse, B.: Climate change impacts model parameter sensitivity – implications for calibration strategy and model diagnostic evaluation, Hydrol Earth Syst Sc, 25, 1307–1332, https://doi.org/10.5194/hess-25-1307-2021, 2021.

Novick, K. A., Ficklin, D. L., Baldocchi, D., Davis, K. J., Ghezzehei, T. A., Konings, A. G., MacBean, N., Raoult, N., Scott, R. L., Shi, Y., Sulman, B. N., and Wood, J. D.: Confronting the water potential information gap, Nat Geosci, 15, 158–164, https://doi.org/10.1038/s41561-022-00909-2, 2022.

Ogden, F. L., Allen, M. B., Lai, W., Zhu, J., Seo, M., Douglas, C. C., and Talbot, C. A.: The soil moisture velocity equation, Journal of Advances in Modeling Earth Systems, 9, 1473–1487, https://doi.org/10.1002/2017ms000931, 2017.

Peters, A., Hohenbrink, T. L., Iden, S. C., Genuchten, M. Th. van, and Durner, W.: Prediction of the absolute hydraulic conductivity function from soil water retention data, Hydrology Earth Syst Sci Discuss, 2023, 1–32, https://doi.org/10.5194/hess-2022-431, 2023.

Samaniego, L., Kumar, R., Thober, S., Rakovec, O., Zink, M., Wanders, N., Eisner, S., Schmied, H. M., Sutanudjaja, E. H., Warrach-Sagi, K., and Attinger, S.: Toward seamless hydrologic predictions across spatial scales, Hydrol Earth Syst Sc, 21, 4323–4346, https://doi.org/10.5194/hess-21-4323-2017, 2017.

Vereecken, H., Amelung, W., Bauke, S. L., Bogena, H., Brüggemann, N., Montzka, C., Vanderborght, J., Bechtold, M., Blöschl, G., Carminati, A., Javaux, M., Konings, A. G., Kusche, J., Neuweiler, I., Or, D., Steele-Dunne, S., Verhoef, A., Young, M., and Zhang, Y.: Soil hydrology in the Earth system, Nat Rev Earth Environ, 3, 573–587, https://doi.org/10.1038/s43017-022-00324-6, 2022.

Vogel, H., Gerke, H. H., Mietrach, R., Zahl, R., and Wöhling, T.: Soil hydraulic conductivity in the state of nonequilibrium, Vadose Zone J, https://doi.org/10.1002/vzj2.20238, 2023.

Wagener, T., Gleeson, T., Coxon, G., Hartmann, A., Howden, N., Pianosi, F., Rahman, M., Rosolem, R., Stein, L., and Woods, R.: On doing hydrology with dragons: Realizing the value of perceptual models and knowledge accumulation, Wiley Interdiscip Rev Water, 8, https://doi.org/10.1002/wat2.1550, 2021.

---

## Author Comment (AC4)

This opinion paper makes interesting and bold claims about the importance of soil properties for hydrology. I agree with many of the statements for natural soils and mature ecosystems. However, the majority of our earth is no longer a natural mature ecosystem. We have changed the surface cover drastically and very large areas are under agriculture or are so badly degraded and not in a mature "steady" state that the ecosystem perspective that is advocated in this paper is possibly no longer applicable. I think that this has to be mentioned in the text and that the reader needs to be reminded more frequently that these statements are made for mature natural ecosystems.

Reply: We thank Anonymous Referee #2's endorsement for the scientific significance of our opinion paper, and the agreement on our statements for natural soils and mature ecosystems. We agree that we are living in a new geological epoch, i.e. the Anthropocene, which means human impacts on essential planetary processes have become profound. Also we agree with the referee's suggestion to say more about the soils in human modified systems, including agriculture, urbanization and deforestation. We did so in the original paper, but we shall bring it out more clearly. However, it is still relevant to emphasize the importance of ecosystem understanding. There are two reasons: 1) also for human modified systems the ecological approach applies, albeit at different and often smaller time scales; 2) the majority of our earth, and particularly the uphill runoff generating parts of catchments, is still dominated by natural ecosystems, although human modification has modified 14.5% or 18.5 M km$^2$ of land (Theobald et al., 2020).

Reference:

Theobald, D. M., Kennedy, C., Chen, B., Oakleaf, J., Baruch-Mordo, S., and Kiesecker, J.: Earth transformed: detailed mapping of global human modification from 1990 to 2017, Earth Syst. Sci. Data, 12, 1953–1972, https://doi.org/10.5194/essd-12-1953-2020, 2020.

That soil is important is clear in situations where severely degraded ecosystems are restored. It is the restoration of the soil that leads to the very large changes in the flow pathways (from overland flow to subsurface flow) and thus streamflow responses. Indeed, it is the ecosystem that changes the soil properties that lead to the changes in the hydrological flow pathways and runoff responses, but this does not mean that the soil itself is not important at all. It means that the ecosystem has such a large effect on soil that the ecosystem would be a better predictor to be used in models (because ecosystem and soil properties become correlated as the ecosystem matures and the ecosystem is easier to observe), but it does not mean that soil is not important at all, especially not when one wants to understand processes. I think that some of the statements about soil not

being important therefore require a bit more nuance. In particular, the model perspective (rather than process perspective) for some of the claims should be made clearer.

Reply: As the referee will have noticed, the first sentence of the paper is: "Soil is important. It forms the substrate of the terrestrial ecosystem and hence is a crucial element of the critical zone of life on Earth". We do agree with the referee and we shall adjust the paper in other locations if this statement is contradicted elsewhere.

One of the confusing parts of the paper is that the authors state that the rooting zone is important but that soil is not important. This seems to suggest that they think that the rooting zone is not part of the soil. I think that what they mean is that soil texture is not important. To me it seems that most of the time when the authors say that soil is not important, they mean that soil texture is not important. For example when the authors refer to soil in the top down approach of catchment comparisons (Section 3.3), they actually refer to texture, not soil hydraulic properties. I urge the authors to more explicitly state that they focus on the soil texture. A better description of what parts of the soil they think are not important would be really helpful. It will also help if they give their definition of soil early in the paper.

Reply: Root zone and soil have strong connections, but with obvious differences. Root zone is the active layer in land surface processes (with as much or even more biomass than above ground) controlling hydrology. Soil is part of the substrate of the ecosystem, but only the root zone is the active part. For example, in the Loess Plateau where soil is thick, only the root zone is the active layer in the topsoil. In Karst and other mountainous regions, rootzone includes not only the soil water storage, but also the fissure water storage in bedrock. In very dry climates, roots can even reach the deep groundwater, thus in this case, the rootzone also includes some part of the groundwater (see Singh et al., 2020). In seasonal cropland, if ploughed, the active part of the soil is limited to the ploughed upper layer and otherwise the rooting depth that a crop can develop within one season. In that case soil properties are indeed dominant. We did mention this in our paper, but we shall make it more clearly since apparently the referee missed that point. For permanent crops, the ecosystem has time to develop its preferred hydrological conditions and our approach applies. If irrigation is provided, human interference comes into play. Also our method has taken irrigation, as extra water supply in dry seasons, into account to estimate root zone storage capacity (Wang-Erlandsson et al., 2016).

Where we talk about soil, we mean soil in general, including soil texture and soil hydraulic properties. Soil texture is widely used in hydrological studies, likely because it is the most easily accessible soil information. Soil itself and its characteristics, such as soil texture and soil hydraulic properties, are the results

of a variety of environmental variables, including climate, base material, topography, and, most importantly, biota. This has been well documented by soil scientists in 19th century, such as Vasily Dokuchaev, one of the most renowned pedologists in history.

Reference:

Singh, C., Wang-Erlandsson, L., Fetzer, I., Rockström, J., and Van der Ent, R., 2020. Rootzone storage capacity reveals drought coping strategies along rainforest-savanna transitions, Environ. Res. Lett. 15 (2020) 124021

Wang-Erlandsson, L., Bastiaanssen, W. G. M., Gao, H., Jaegermeyr, J., Senay, G. B., van Dijk, A. I. J. M., Guerschman, J. P., Keys, P. W., Gordon, L. J., and Savenije, H. H. G.: Global root zone storage capacity from satellite-based evaporation, Hydrol. Earth Syst. Sci., 20, 1459–1481, https://doi.org/10.5194/hess-20-1459-2016, 2016.

The authors should point out much more clearly (and explicitly) that a major problem is that we use texture in pedotransfer functions to derive the soil characteristics that are related to water flow and storage, especially because these pedotransfer functions were developed for agricultural soils. The sand or silt content of a soil do not affect water flow or storage. We only attribute such an effect when we use pedotransfer functions to derive properties related to water flow and storage based on the texture. Because the pedotransfer functions were largely derived for agricultural soils, they do not take the effects of structure (and preferential flow) into account.

Reply: We agree completely with this comment and will follow up on this suggestion

The writing of the manuscript could be a bit sharper. At several places, the authors make a good argument for why the ecosystem is important and then conclude that the soil is not important. I think that these sections need to be improved for two reasons. First, reasons are given for why the ecosystem is important but not for why the soil is not important. In particular, no references are given for this second part. In other words, the authors provide arguments for the first part (the ecosystem is important) but not for the second part (soil is not important). Thus either the second part (soil is not important) has to be taken out or arguments and references need to be included for the second part as well. Second, ecosystem and soil are interconnected. It is the ecosystem that changes the soil properties. So one can not directly argue that because the ecosystem is important, the soil is not important. It is still important but the ecosystem is perhaps the better predicting variable to be used in models because it is easier to observe and has a large effect on the soil properties that actually affect how water moves through the soil.

Reply: Thank you for pointing this out. We will improve the writing accordingly.

Other parts of the writing could also be improved. In several sentences words are missing and some other sentences are not clear and should be reformulated. The structure of the paper and individual sections was sometimes unclear to me. For example, section 4.1 consists of four paragraphs. Paragraph one highlights the importance of ET and states that hydrologists focus on discharge instead (but this point was already made on L128). The second paragraph then describes that ecosystems maximize storage and drainage. This section is interesting and fits the caption of this section. One would expect the next paragraph to get deeper into this but the third paragraph describes that the numbers for soil properties used in models don't match the actual measurement values, and the fourth paragraph describes the rebalancing of soil properties that needs to be done in models. While the first two paragraphs sort of fit together and the last two paragraphs as well, the link between the first two and last two is not obvious. It also means that the second paragraph ends abruptly and this line of thinking could use some more elaboration. In addition, the part on the soil properties and the rebalancing starts abruptly without an introduction. The latter two paragraphs would probably better fit in a separate section on the problematic part of using pedotransfer functions based on texture (see comments above). This is just one example, there are other sections where the flow was unclear and I expect other readers to also wonder how the paragraphs are connected. I made some suggestions in the annotated pdf but there are more places where text could be reordered for a better flow. I don't request that the authors use the suggested order but I do recommend that they carefully read through the paper to see if the order is logical for a reader.

Reply: Thank you for pointing this out. We will make the suggested changes to make them more logical for the readers.

Oter specific points:

- L56/139: I think that the problematic part of the use of pedotransfer functions based on texture to derive properties about pores should be described in more detail. Especially knowing that these pedotransfer functions were developed based on cores from agricultural fields and that texture does not really influence the hydraulic concucitivity (e.g., Jarvis et al., 2013; Gupta et al., 2021). See also comments above.
- Reply: Thank you. We will follow up on your suggestion and also add relevant references in the revised MS.
- Section 3.1: I don't think that anyone claims that soil affects the long term water balance more than climate and vegetation. So, I think that it is fine to use

this section to highlight that the ecosystem and climate are the main factors that determine the long term water balance but it makes less sense to use this as an argument that soils are not important.

- Reply: The logic is we separately discussed the role of soil in both long term water balance and short term hydrological processes. We believe it is relevant to clarify the unimportance of soil for the long term water balance, but we shall not overemphasize this point.

- L129-132: Yes, land use change (if severe) alters runoff generation, exactly because of the large effect it has on soils. So, I don't think that you can use this argument here to say that soils don't matter. You can use it to make the argument that vegetation has a large effect on the soil properties that actually matter for water flow and storage. Also, it would be good to reference some field studies here (not only model studies).

  Reply: We will follow your excellent suggestion to rephrase this sentence and add more references about filed studies.

- L158: But the comparison is basically between a model and a model with more data. I don't think that one should call this observations.

- Reply: Thank you for pointing out this issue. We will change the term "observed" by "Remote Sensing derived".

- L162: But it also mimics the depth to the groundwater – maybe this has a different effect in the two models?

- Reply: Yes, the depth to the groundwater also impacts the evaporation in dry seasons in the Netherlands. The soil-based model heavily relies on detailed soil observations, which did not consider groundwater replenishment, and underestimated evaporation.

- L181: The problem is in part that we use texture here. Texture does not describe the soil pores that are important for storage or flow of water. The problem is that we use pedotransfer functions that are largely based on data from agricultural soils and are not appropriate for forested systems. See also the comments above. Furthermore, soil depth data is usually very rough and not very reliable. Maps of soil properties that actually describe water flow and storage are rarely available. Thus, one could also argue that the big problem is that we don't have soil maps with sufficient information on the properties that actually matter and are related to water flow, and that instead we rely too much on texture and pedotransfer functions.

- Reply: Yes, it is a technical issue to rely too much on soil texture and pedotransfer functions. The more fundamental issue though is whether we understand and model hydrological processes based on the substrate (the soil) or on the active agent (the ecosystems). It is an issue of cause and effect. We have focused too much on the effect (the soil properties that we observe locally

but cannot observe at the relevant scale) instead of trying to understand the agent that creates the soil properties, which acts at the appropriate and observable ecosystem scale.

- L188: I agree that all these processes are intertwined or connected. Therefore, I think that the opinion paper should use more careful wording. It is OK to say that for hydrological modeling it is more useful to look at the ecosystem because the soil properties that matter for hydrology are highly correlated with land cover, and ecosystem properties are much easier to observe or measure. However, if we want to actually understand processes and the factors that affect these processes, it is important to look at the processes. In other words, then we have to look at the partitioning of rainfall into infiltration, overland flow, deeper drainage, etc. and soils are important. I think that this distinction between model application and process understanding should be made more clearly throughout the text.

Reply: Our opinion paper discussed the role of soil in both model development and process understanding. We found that soil is overrated, not only in model development but also in process understanding. The role of soil is overrated not only in catchment hydrology, but also in hillslope runoff generation under natural condition, land surface evaporation and energy interaction. Even small-scale water movement and pathways are not mainly driven by soil properties but by soil structure, controlled mostly by the ecosystem. Moreover, our argument is about what is the active manager and main driving force, and what is the substrate? What is the dependent variable and what the independent? What is cause, and what is consequence? And eventually what is intuition, and what is realism?

We also believe, and it this is hard to prove as yet, that partitioning is controlled by the ecosystem, firstly by interception and throughfall concentration in dripping points where infiltration is facilitated, next by preferential infiltration patterns that are created by biota, and third by subsurface drainage and percolation. From an evolutionary perspective it is reasonable to assume that ecosystems evolve towards survival (a Darwinian hypothesis). This implies that surface runoff is prevented (causing loss of nutrients and fertile topsoil), that depleted moisture stocks are quickly replenished, and that excess water is drained below the root zone (rapid subsurface flow). From a larger ecosystem perspective, one could even go as far as assuming that recharge of groundwater is beneficial to the ecosystem at larger scale, sustaining base flow. We have not even touched upon all the intricacies of how ecosystems manipulate the substrate to its advantage. We have merely shown convincingly (by several model applications) that the root zone storage that ecosystems create are better predictors of hydrological behaviour than soil texture derived storage. In this opinion paper we do not claim that we have all the knowledge required to explain partitioning, we merely point a more promising research direction to untangle hydrological complexity, where the basic assumption is that an active agent, with a clear purpose, creates its own

conditions for survival. As a bonus, models based on this approach appear to be simpler, cheaper, less time and resource demanding and better at the job for which they are developed, see for instance Mao and Liu (2019).

Mao, G. and Liu, J.: WAYS v1: a hydrological model for root zone water storage simulation on a global scale, Geosci. Model Dev., 12, 5267–5289, https://doi.org/10.5194/gmd-12-5267-2019, 2019.

- Section 4.2: I am sorry but I don't understand what these ERA5 storage volumes contribute to the arguments of the opinion paper. The volume is one thing, the total flux from repeated filling and emptying is another. Certainly, I agree that the total storage is highest in the root zone but I consider the root zone to be part of the soil. So why is the root zone important but soil not? The paragraph on 277-284 goes some way into explaining this but it could have been added to section 4.1. It would be good if the authors give a definition of soil early in the paper. I have the feeling that often the authors mean soil texture instead of the soil itself.

- Reply: For the relationship between soil and root zone, the Anonymous Referee #2 can find our replies to your main comment. We will add the definition of soil in the revised MS.

- Several minor comments and suggestions are given in the annotated pdf.

Reply: We thank Anonymous Referee #2's comments are suggestions which are greatly helpful to improve the quality of this manuscript.

References:

Jarvis, N., Koestel, J., Messing, I., Moeys, J., and Lindahl, A.: Influence of soil, land use and climatic factors on the hydraulic conductivity of soil, Hydrol. Earth Syst. Sci., 17, 5185–5195, https://doi.org/10.5194/hess-17-5185-2013, 2013.

Gupta, S., Hengl, T., Lehmann, P., Bonetti, S., and Or, D.: SoilKsatDB: global database of soil saturated hydraulic conductivity measurements for geoscience applications, Earth Syst. Sci. Data, 13, 1593–1612, https://doi.org/10.5194/essd-13-1593-2021, 2021.

---

## Author Response (AR1)

Dear Dr. Gao,

after reading your interesting opinion, the reviews and scientific comments and your response, I was excited about the quality of the debate, which is beyond my expectations. So the generic purpose of this opinion paper is more than fulfilled and your work is definitely worth to be published in HESS. Yet I think that the revised version of your paper needs to properly pick up and discuss key points raised by the reviewers and the commenters.

**Reply**: We thank the Editor's endorsement for the scientific contribution of our opinion paper and the quality of the debate during discussion. We are also grateful to all four reviewers' for their debate and constructive comments, which have been greatly helpful to improve the quality of the manuscript. The manuscript itself now has been further sharpened, and thoroughly improved based on their valuable suggestions.

We've made detailed point-to-point replies during the discussion already. The main concern from both Dr. Teuling and Dr. Brocca is that the term "observed" evaporation in Figure 1 implies too high a claim of truth. We changed the term "observed" by "Remote Sensing derived" in the revised version. Hence, here we'd like to not repeat our reply. In this round of revision, we provided a complete answer to all the questions from Conrad Jackisch and Anonymous Referee #2, and pointed out the actual changes in the revised paper.

Dr. Teuling made a valid point that a mismatch with satellite-based evaporation retrievals, does not necessarily speak against a model, because the former can be of poor quality as well. While I appreciate your response here, I am not sure whether Dr. Teuling refers to "point" measurements in his comment, as e.g. flux tower data have a footprint of 1 km2.

**Reply**: We changed the term "observed" by "Remote Sensing derived" in the revised version.

Dr. Jackisch stressed, that there are many efforts to step beyond the Richards equation and to account for preferential flow and non-equilibrium infiltration. One Lagrangan approach has been developed by Dr. Jackisch and myself (HESS 2018, HESS 2016), and further expanded by and successfully tested against tracer and even pesticide data, that revealed strongly preferential transport Sternagel et al (HESS 2019, HESS 2021). The approach does not rely on the Darcy equation, can easily account for imperfect mixing and preferential flow, and it can be easily up-scaled. It does however account for capillarity in soil, and you surely agree that storage against gravity is controlled by capillary forces.

**Reply**: We cited more new development in preferential flow modeling studies.

Also the storage against gravity is discussed in the revised Introduction.

I agree that the ecosystem is key (but only in pristine mature areas, as pointed out by reviewer 2) and the partially saturated zone does not work as a simple, porous filter. I'll also agree that soil texture is, as also pointed out by Dr. Jackisch and Reviewer 2, a very poor representation of the "soil ecosystem" in our minds and model. Yet I wonder whether it might help to acknowledge:
- The soil is an essential part of the ecosystem as it provides the storage volume (no storage without a storage volume, no evaporation without storage). The soil provides also nutrients for plant growth and it is habitat for soil fauna. Plants change infiltration and drainage properties, so do earthworms. Overall, I agree with Dr. Jackisch that the concept of soil type, which is a generic and holistic one, represents the soils as a whole much better.

Reply: The role of soil in ecosystem is discussed more in the Introduction and Section 4.1 on ecosystem hierarchy. We added Figure 4 to demonstrate our hierarchy perspective that soil is an essential part of the ecosystem.

- The concept of capillarity is much more universal than the Darcy- Richards approach. It relates the high surface tension of water and to the porous nature of the subsurface (providing a storage volume). But there wouldn't be any storage against gravity in the pore space, without capillarity. As soon as a conceptual model uses a soil/root zone moisture accounting, it implicitly relies on capillarity in soils.
Reply: We agree. The concept of capillarity is added in the first paragraph of the main text.

- While capillarity is important for storage, I agree that it is not important for infiltration, recharge and subsurface runoff generation, all this is preferential as you correctly pointed out in your Opinion. Maybe it would be helpful to distinguish wetting and drainage, because they relate to different forms of preferential flow, different subsurface structures and have different benefits for the ecosystem.
Reply: We agree. The wetting and drainage have different benefits for the ecosystem. They are distinguished and discussed mostly in Section 4.2.

- I like the idea very much that the necessary root zone storage for an ecosystem to survive a drought determines rooting depth. But is this not independent of the retention function, otherwise the plants need to tab groundwater (and root therein)?
Reply: Rootzone storage and soil have a strong connection, but are essentially two different things. We discussed their complex relationships in different climatic and lithologic conditions in the end of Section 4.3.

Looking forward to receive the revised manuscript,

Erwin Zehe

**'Critical comments on egusphere-2023-125', Conrad Jackisch**

Hongkai Gao and co-workers present an interesting contribution to the debate about key elements in the concepts of hydrological modelling. As an opinion paper, the authors argue that the affinity of hydrological model concepts to soil properties are more a relict than a substantial information basis. They propose to shift focus to the rootzone (as manifestation of the ecosystem) as alternative conceptual foundation.

**RC1.1:** I congratulate the authors for their work and I agree that our community has to keep challenging the conceptual assumptions and traditions. The role of soils in hydrology and land surface modelling is a particularly interesting debate. Recently Novick et al. (2022) have pointed to a "water potential information gap" in similar notion but opposite proposals. Discussing the role of pedotransfer functions (Looy et al. 2017, Vereecken et al. 2022) and soil hydraulic functions (Peters et al. 2023) together with structural adequacy of models (Gupta et al. 2012), perceptual model consistency (Wagener et al. 2021) and the data flow in model building (Gharari et al. 2021) and model analyses (Loritz et al. 2018) is in my view very important and promising. Hence I see the topic of this manuscript as worth an opinion paper.

We thank Conrad Jackisch for endorsing the need of our work, and his pointing out the relevance of this topic in relation to various developments in hydrological science and land surface modelling.

  **RC1.2:** However, I am not really convinced that the current arrangement of the arguments in the manuscript is really substantiating this timely debate. My main concern is that the authors use the word "soil" for different concepts and at different scales without much differentiation. The critical zone concept (Lin et al. 2006) was already much further than this. Also the debate about landscape organisation and hydrologic functioning (Jackisch et al. 2021) including a critical assessment of conceptual assumptions about processes and scaling is more advanced on the topic.

We thank the reviewer for referring to previous work that is relevant to our discussion, and we have included these references in our revised version.

The difference between our ecosystem centred approach and critical zone concept can be found in Section 4.1, especially Figure 4, and the last paragraph in Section 4.3.

**RC1.3:** The authors do address such aspects in their manuscript and point to

intertwined factors and sub-systems. However, the arguments are not really brought to consistently support the very fundamental claim of the manuscript. Without meaning to offend the authors, I would see many of the claims rather being rooted in conceptual limitations in the view of soil functions by the authors than in the lack of information or importance of soils in hydrological processes and models. I will substantiate this in the more detailed assessment.

We do not agree that our abandoning detailed description of soil processes is caused by a lack of understanding of such processes. The limitations of this perspective can be found in Section 2 and Section 2.

Moreover, detailed small-scale soil knowledge does not prevent us from seeing the larger scale picture. Our point of view is summarized in "Section 4.1 Ecosystem hierarchy" and Figure 4.

**RC1.4:** In general, I do not really see, how the replacement of a soil-centred with a rootzone-centred concept deviated from the critical zone concepts (Lin et al. 2006). I also do not see, why the authors omit the main driving concept for fluxes (depletion of gradients) and thus the whole debate about potentials (Novick et al. 2022). I would have liked to see to which degree their arguments are essentially an expression of the conceptualisation of hydrological models i) as distributed and linked storages, ii) at a broader scale (in the sense of the scale triplet) and iii) with soils expressed by texture classes.

The difference between our ecosystem centred approach and critical zone concept can be found in Section 4.1, and the last paragraph in Section 4.3.

Novick et al. 2022 is about potentials that govern the water flows throughout the soil–plant–atmosphere. These processes are much more specific than the level of detail we are focusing on in our commentary which is at larger scale hydrological behaviour.

In order to avoid these misunderstandings, we clarified the limitations of our discussion in our revised version that we are primarily interested in the description of hydrological processes at system scale (in Section 6).

**RC1.5:** Moreover, I find many claims very strong and confrontative (e.g. L21f, L111) and not well-balanced. To really spark the debate (and not a battle) I would have liked a more balanced and substantiated formulation.

To make our arguments more balanced, we added Section 6 clarifying that our ecosystem approach is also subjected to certain limitations.

Detailed comments:

**RC1.6:** L43: I would argue that this is the debate throughout in pedology. At least not only recently.

"Soil forms an ecosystem in itself" is a common sense in pedology. This can be found in many soil textbooks, such as "The Nature and Properties of Soils. 15th edition, Weil and Brady, 2017". As suggested by the reviewer, we've removed "recently".

**RC1.7:** L49: Why do you limit the perspective to abiotic boundary conditions when you actually argure for an ecosystem perspective? Biodiversity, niches, disturbances, stressors, carbon pools are not all determined by climate and geology. Moreover, at least most temperate soils do not develop directly from bedrock material but on deposited material from rather old geomorphological processes (which include path-dependent development options). Pointing to this, soil degradation and soil loss too is an important and largely irreversible process with severe implications on regional hydrological and biogeochemical cycles.

We've rephrased this sentence, which can be found in Line 44-48.

**RC1.8:** L59: I think I have an idea what you intend to express, but since this simplistic/reductionist pedotransfer approach has a couple of implications which could be challenged. I suggest to clarify this sentence a little more and link to the debates in soil physics and the pedotransfer community.

We've clarified this sentence more in Line 69-70.

**RC1.9:** L63 (opposite): I am not sure if I can follow. The argument before was that simple pedotransfer and soil hydraulic property models are an issue and that they become coupled to rooting depth.

Consequently, rooting depth has to be defined somehow to eventually assess plant-available soil water storage. Why does it matter if in this step plant-available soil water storage becomes the dependent variable of the other involved variables, if it is used as independent variable in the proceeding calculation steps? Isn't this a question about the perceptual model underlying any form of conceptualisation and numerical expression?

L63 (root zone storage): Yes, but maybe at different time scales? Plants and ecosystem may adapt and co-evolve (within a range of their survival). So why should the debate be solved by exchanging the depending/independent variables? Could this actually be a scaling issue?

From a soil-based perspective, the rootzone storage is commonly estimated as a function of plant available moisture and rooting depth (Yang et al., 2016). In our view, this approach is not satisfactory, as it considers plant available moisture and

rooting depth as independent variables, and rootzone storage as the dependent variable. We argue the reverse: plant available moisture and rooting depth are a function of the rootzone storage that is created by the ecosystem to fulfil its water management strategies. Moreover, the classical approach is impractical, as obtaining the detailed spatio-temporal root and soil information at a global scale is virtually impossible (Or, 2020).

The details can be found in L362-369, and in Section 4.1.

References:

Or, D.: The tyranny of small scales–On representing soil processes in global land surface models, Water Resour. Res., 56, 1–9, https://doi.org/10.1029/2019WR024846, 2020.

Yang, Y., Donohue, R.J., and McVicar, T.R.: Global estimation of effective plant rooting depth: Implications for hydrological modeling, Water Resour. Res., 52, 8260–8276, https://doi.org/10.1002/2016WR019392, 2016.

**RC1.10:** L65: This might depend on what exactly we see as detailed. As you open your argumentation with coevolution, maybe a broad idea about the general type of soil (not texture class) and biome (including its ecohydrological properties) could be sufficiently detailed? If so, remote sensing claims various solutions to gather such data…

We moved this sentence to Section 4.3, with more detailed discussion about the difference between classical soil-based approach and rootzone perspective. Please see Line 362-369.

**RC1.11:** L67: Again, i would see this as a scale issue: Ecosystem and climate are both terms referring to large scales (in the sense of the scale triplet). Hydrology is not referring to a specific scale.

We agree that we are referring to a larger scale. The scale triplet referred to by the reviewer is the 'Process scale', 'observation scale' and 'modelling (working) scale' defined by Bloeschl and Sivapalan (1995). As clarified earlier, we are considering processes at the system scale. This is also our modelling scale, as we are interested in characterizing the processes directly at this scale. We've clarified the scale issue in Section 4.4 and certain limitations of our ecosystem approach in Section 6.

**RC1.12:** L72f: This assumes that the ecosystem is somewhat in equilibrium with determinable drivers of its development. However, path-dependent trajectories, dynamic deviations from equilibrium more or less buffered by the ecosystem and any application for global changes (climate, land use, cohabitation…) but severe challenges to this view.

We removed this sentence in the revised version. We did more discussion on the common practice of using soil characteristics and rooting depth in hydrological modelling in Line 362-369. Regarding the challenges from global changes (climate, land use and other human drivers), we discussed these in the last paragraph of Section 5.1, and Line 492-494.

**RC1.13:** L78: Why do you refer exactly to these citations? I would think that e.g. the work of Gardner including his famous lab experiments have been far more important for propagating this perception.

We changed the citations.

**RC1.14:** L87ff: Yes, and this might be one of the actual issues to address here. Linear Darcy filter flow has been coupled to highly non-linear retention properties with the Richards equation and as a first-order diffusive flow model, it does an ok job for diffusive flow in somewhat well-defined porous media. However, especially infiltration (as initial soil water redistribution into the soil during rain events) is often not dominated by diffusive flow but by advection (Newtion Shear Flow equation (Germann, 2020), soil moisture velocity equation (Ogden et al. 2017), particle model (Jackisch and Zehe 2018), non-equilibrium flow (Vogel et al. 2023)). To my understanding, this deficit is rooting back to the very limited means to measure antecedent state-dependent infiltration and to use such data in hydrologic models. But why this is an argument for soils not being central in the question for one of their fundamental services to mediate the local soil water cycle is not clear to me. Especially because infiltration is state dependent, precipitation may not be retained after drought conditions, requiring vast amounts of light rains, slow snow melts or similar to replenish the water stocks, while storm events will simply lead to preferential flow and possibly erosion…

We agree that soil fulfils several important functions. Provided that we are considering ecosystem-scale processes, and therefore integrating processes that take a minimum area to be operative, the question is whether processes such as infiltration, retention or release to subsurface flow depend on soil properties such as texture or can be somehow related to them, as many hydrological models assume. Our suggestion is that the processes that are significant at the ecosystem-scale are conditioned on a multitude of soil properties, such as macropores, rootzone depth, etc, which are themselves conditioned on the vegetation, which ultimately is conditioned by climate. In order to model the ecosystem-scale hydrological processes, in this set of nested dependencies, it is sufficient to stop at the ecosystem level. There are other applications, however, where it is necessary to dig into the soil level, as shown in Figure 4 and Section 4.1.

**RC1.15:** L93: Well, it is not dominant when it comes to storm events, yes. But these experiments use rather steep gradients with a lot of water. The debate

about when and to what degree soil water flow is preferential is ongoing. If this was the full story, soils could hardly sustain the ecosystems.

We agree that "The debate about when and to what degree soil water flow is preferential is ongoing". Here we merely focused on tracer field experiments, which have many literatures and documentations. We made changes in Line 118-119.

**RC1.16:** L98: Partly yes. But preferential flow can also simplify our models. Anyways, I suggest to ease the dispute opened by this statement with a slightly more balanced view on achievements towards unifying forms of non-uniform infiltration.

We removed this sentence.

**RC1.17:** L108: Yes. But this again can be seen as a scaling issue. At the hillslope- and plot-scale, these parameters/concepts have been very central. Only at the catchment-scale they could be easily subsumed as general soil property parameters not requiring for a dual domain definition. And this is true for the hindcast of our observations…

We clarified the scale issue in this opinion paper in Section 6.

**RC1.18:** L111: Again a strong claim. I can agree that the pref flow debate has always struggled to connect to Darcy-scale soil physics. But fog? No progress? Is this claim really needed for your argument?

We removed this sentence in the revised version.

**RC1.19:** L113 (a priori assumption): Well, they are ABOUT the description of soil water flow. If they are key for describing hydrological processes is part of the actual model conceptualisation, its numerics and the respective regimes under study. Again, I would argue that this is no other "a priori assumption" as most other parts of the perceptual model. And since its actual effect in the model can be and is challenged (Glaser et al. 2018), I would rather see it as a positive example for advancing hydrologic models.

Yes, we agree that all our perceptual models need experiments to test and sharpen. Both accepting and rejecting a-priori assumptions is good for advancing our understanding of the hydrological system.

**RC1.20:** L118: So far soil variability has not been motivated. This is especially difficult, because the effect of soil variability is again a matter of scale (including the respective range of processes). After reading subsection 3.1, I can think of quite a number of papers, providing good evidence for the opposite: When you have the average soil right, you can easily reproduce observed hydrologic patterns (e.g. Loritz et al. 2017).

As mentioned above, we clarified the scale issue in Section 6.

**RC1.21:** L123f: Ok. Known and well established. Maybe citing some of the many studies would be nice.

We removed the entire section on long-term water balance.

**RC1.22:** L131f: This argument is not really sound. Studies fully agree that plants and ecosystems strongly moderate the net ET flux of a stand. But without soil as the part of the ecosystem which can actually store water for weeks and beyond, this percentage cannot be reached. We exactly see this in data based on Budyko-like assessments that more draining locations (sandy, karstic) have very little ETact simply because precipitation is largely drained.

We removed the entire section on long-term water balance.

**RC1.23:** L137f: Yes, difficult but steeply advancing. Please see Peters et al. (2023) and Hohenbrink et al. (submitted to ESSD) for examples. The most critical part might be the reduction of such data to van Genuchten/Mualem SHP model parameters and the weakly informative relation to the broad texture classes, BD and Corg. But the issue of pedotransfer models is a discussion on its own, and which is currently gaining momentum.

We cited more recent papers, as the reviewer suggested.

**RC1.24:** L144: The issue here might be that soil mapping is not particularly done for hydrological purposes. On the one hand, pedological classes are not always directly convertible to hydrological properties. On the other hand, soil stratification and the respective hydrological properties are rarely conveyed into land surface models with sufficient degree of vertical resolution. Moreover, the uncertainty about the hydrological properties of the mapped soil classes is largely unknown and very different from region to region. Given all of these points, I am not quite sure if "interpolation and upscaling" is the core issue here. Maybe it is more a disconnection between soil mappers and hydrologic modellers?

We agree that soil maps are not particularly useful for hydrology. There have been attempts to develop hydrological soil maps such as 'the hydrology of soil types' (Boorman et al, 1995) in UK, but these are not widely available. However, our main argument is that pursuing that route is unnecessary if one is interested in describing catchment scale processes (see section 4.1). This does not mean that a connection between soil mappers and hydrological modelers would not be fruitful, particularly if the focus is on describing smaller scale processes (see Section 6).

**RC1.25:** L148: I fully agree that unnatural lab conditions are a fundamental difficulty. However, many measurements are conducted using "undisturbed"

samples for soil hydraulic property analyses. It is unnatural because the samples are extracted from their capillary context, exposed to free evaporation at the surface and a no flow boundary at the bottom (for the standard HYPROP protocol). However pedotransfer functions are then correlating lab measurements (soil hydraulic properties) to lab measurements (texture) and the scaling and transfer involved in its application to field conditions remain hidden.

We agree that with a lot of effort one can obtain reliable pedotransfer functions, however, this is impractical at the catchment scale. Moreover, such functions are typically surface maps, and lack a vertical dimension, which is necessary to model what happens in the underground. The pedotransfer functions approach is plagued with uncertainties and difficulties of various kinds. For this reason, we believe that exploring alternative, potentially easier, approaches is worthwhile.

**RC1.26:** L152: I fully agree, but again this is an issue with quite a bit of literature from hydropedology to cite here.

We added more citations from hydropedology community in the revised version.

**RC1.27:** L153ff: I do not get this point. The discussion about parameter regionalisation has a long standing in hydrology. E.g. mHM (Samaniego et al. 2017) exactly works because it modifies the initial lab scale parameters to match its distributed effects on fluxes in the landscape. Showing one odd model result can have so many reasons that I find it very difficult to support your argument through it.

The Netherlands likely benefits from the most detailed soil surveys in the world, thanks to its very advanced agricultural science and technology. But even with such detailed soil data, the soil-based evaporation model produced such large discrepancy. This is a strong indication that collecting detailed soil data does not benefit hydrological studies. This is not one odd model result, but a strong example showing the dead-end track of this methodology.

**RC1.28:** L157: Which is a nice example for model extrapolation and the shift in parameter sensitivity under climate change (Melsen and Guse 2021).

Thank you for sharing this literature. We have thoroughly improved citation in the revised version.

**RC1.29:** L177: Again, a difficult claim. They test if texture classes and soil depth is informative. However, Novick et al. (2022) point nicely to soil water potential being most informative and often omitted in LSMs. The model you are referring to are not particularly strong in soil physics as they conceptualise soils as stores instead of any framework of potentials as drivers. So your assessment might actually pinpoint that soil hydrology based on a storage concept is not very informative? As stated in the general section, I find it very difficult that you do not

discern between weak conceptualisations of soils and the actual physical properties and dynamics linked to soils.

Please refer to our reply to RC1.4.

**RC1.30:** L188: These intertwined factors mostly manifest at "soil scales", which are not necessarily very small.

Our attitude to terminology in this opinion paper can be found in RC1.13.

**RC1.31:** L194f: Again, I would argue that the concept of infiltration capacity as rigid site property maybe the root of the issue here? Infiltration capacity to my understanding does not necessarily entail a constant or any specific model (e.g. Horton which is subsuming site properties and antecedent condition into an exponential decay function for infiltration rate or Green and Ampt which indeed is rarely proven in natural soils). Since infiltration is the passage of water into the soil domain, I would argue that soil structures (draining macropores and storing finer pores) facilitate it and that antecedent conditions plus the rainfall supply dynamics govern the individual initial (non-uniform) soil water redistribution (see comment to L87ff). The ecosystem modifies the boundary conditions, state dynamics and structure formation in the long run (Lange et al. 2015 and other publications from the Jena experiment).

The reviewer's thought on infiltration capacity is interesting. We agree that "infiltration capacity… does not necessarily entail a constant or any specific model", which is in line with our augment to question the long-term held belief that soil determines infiltration capacity. The concept of infiltration capacity is still important in hydrology, especially for storm events. But in natural hillslope and catchments, vegetation, topography and land surface are all indispensable factors in storm event modelling and may play a more significant role controlling the infiltration capacity than soil properties (see Line 309-312, and the new Section 4.5).

**RC1.32:** L203ff: I agree and I admire the authors for their very nice contributions to these examples. However, this comparison is not fair since the intended applications of more complex models are often more than rainfall-runoff modelling. Especially when models are used to analyse effects of changes in land use , climate regime, management etc. the stationarity assumption collapses and we require parameters and submodels with physical meaning. Once we have a good understanding about how the modified hydrologic system can be conceptualised, the simple models are much more efficient and maybe even less error-prone again. But the transition (in system characteristics or scale) remains very challenging for these kind of models.

The reviewer might not get our points. We believe the reviewer is also with us. Both of us agree that we should not develop our model based on stationary

assumption. But the soil-based model is a typical stationary model, since soil properties are mostly stable and unchanged with climate and human activities in the short term. What changes dramatically are the land use and land cover and belowground biomass in the background of both human activities and climate change. Our proposed ecosystem-based approach is not only essential for hydrological understanding in natural ecosystems, but also deals with this human-impact issue much better than soil-based models. Because an ecosystem-based model intrinsically regards a catchment as a living organism. The reviewer may find our discussion in Line 471-478, Line 492-494, and refer to many published papers (Nijzink et al., 2016; de Boer-Euser et al., 2019; Bouaziz et al., 2022).

**RC1.33:** L210f and Fig. 2: I do not find it a logical proof of your argument that some models can succeed without soil information. If soil information is only texture class and porosity maybe it is more telling that these properties are not very informative for hydropedological characteristics and that the variable for the most frequent antecedent conditions (aridity) has far more influence because it is more informative for hydrological functioning? Hohenbrink et al. (submitted to ESSD, soon at https://doi.org/10.5194/essd-2023-74) show very nicely how these standard properties and texture-based soil classes do not inform hydropedologic functioning.

Soil texture is the most easily accessible soil information, that could be the reason Addor et al (2018) chose these characteristics to compare with hydrological signatures.

**RC1.34:** L217ff: I find it difficult to discern your "ecosystem"/"rootzone" approach from the hydropedology concepts (Lin et al. 2006).

Root zone and critical zone have strong connections, but with obvious differences. The lower and upper boundary of the critical zone is still debated. But usually, Earth's critical zone includes air, soil, water, substrate rock and organisms. For hydrology, the ecosystem with its rootzone is the most active layer in the critical zone. For example, in the Loess Plateau where soil is thick, the root zone is merely the active layer on the topsoil. In Karst and other mountainous regions, the rootzone moisture storage includes not only the soil water storage, but also the fissure water storage in the bedrock. In very dry climates, roots can even reach the deep groundwater, thus in this case, the rootzone also includes some part of the groundwater (see Singh et al., 2020). In cropland, where irrigation provides an extra water supply to rootzone during dry seasons, the rootzone water storage capacity is often smaller than under natural conditions under similar climate conditions. The rootzone is the most active layer in the critical zone (with as much or even more biomass than above ground) controlling land surface processes, including hydrology. The reviewer can find our detailed discussion in the last paragraph of Section 4.3.

Singh, C., Wang-Erlandsson, L., Fetzer, I., Rockström, J., and Van der Ent, R., 2020. Rootzone storage capacity reveals drought coping strategies along rainforest-savanna transitions, Environ. Res. Lett. 15 (2020) 124021

**RC1.35:** L226ff: Within the lines of arguments, I think you are jumping through different scales here (with concepts and properties which are known NOT to be scale-invariant). The assumption that the ecosystem will be able to become the dominant driver is only true if the system has sufficient degrees of freedom to do so. Mediterranean basins have been deforested long ago, soil has been lost and there is no sign of spontaneous ecosystem replenishing under the current climate conditions. Badlands, crusts, long-term unstable debris are examples contradicting your claim. Hence a more differentiated analysis would be more insightful?

The scale issue we discussed in this manuscript is clarified in Section 6.

**RC1.36:** L232: I fully agree that water can bypass the rootzone but is not necessarily reaching groundwater. In many soil systems of the mid latitudes we find laterally conductive layers formed by more distant ice ages leading to relatively quick drainage or even interflow. Your FLEX approach has nicely shown this for the Ardennes…

We added subsurface storm flow in Line 327-328.

**RC1.37:** L235ff: With having FLEX in mind I can understand your reasoning but I find your PERCEPTUAL model rather inflexible in the first place. The notion to simplify as much as possible is fully legit but deterministic concepts are in my understanding rather a thing of the past when we were limited in computational powers. And I find that this stiffness weakens your argumentation.

We thoroughly improved the entire Section 4.2.

**RC1.38:** L245 (and the paragraphs before and after): I do not see why this is an argument against the importance of soils. Just because modellers use non-informative variables about soils and just because they have not found laws to scale the scale-dependent concepts/models does not mean that soils are not important. If these observations are biased, this does actually point to a misconception of the soil system rather than serving as an argument for omitting soils altogether. I would claim that this only shows that soil function cannot be described by texture classes (alone).

It is worthwhile to note again that we did not intend to omit soils altogether. We claimed that "Soil is important" at the very beginning of our paper. We proposed to considering root zone as an integrated system, rather than simply treating soil and roots as isolated parts.

As the reviewers suggested, this paragraph may not be properly placed here, we moved it to Section 2.2.

**RC1.39:** L251: I find it very difficult to agree to your arguments at this too general level of characterisation of somewhat arbitrarily selected model examples. I suggest to build the arguments based on the state of the art about structural adequacy and model conceptualisation (see general comments)

Please see our response to your general comments.

**RC1.40:** L282ff (and the whole subsection): You are proposing a new conceptualisation in which you omit various central properties governing water retention and drainage, which are not only governed by vegetation alone. With most of the terrestrial surface of our planet being actively managed by humans and a massively changing climate and biosphere, I find it not very helpful from a physical and system perspective. Moreover, your concept does not evade the scale issues. Quite to the contrary the active rootzone is not a static thing (at many scales). When we look at root water uptake alone, the sourcing depth of water within the root zone is dynamic over the year and very different from site to site (with the very same tree species and ages) (Jackisch et al. 2020). Giving reference to ERA5 data for this is maybe a little too large scale to substantiate your arguments with?

We agree that the active rootzone is not a static thing (at many scales), thus we need to develop an alive model to take these changes into account. This is exactly what we are saying in our opinion paper. Please refer to some of more references (Nijzink et al., 2016; de Boer-Euser et al., 2019; Bouaziz et al., 2022).

Our argument refers to the catchment scale. The concept of preferential flow was proposed in small scale soil profiles, but hydrologists found that preferential flow is everywhere for all hydrological processes at multi-scales (Uhlenbrook, 2006). Also, the root zone is not only important for catchment hydrology, but also for land surface processes, and is essential for ecosystem's resilience to drought at multiple scales, including landscape, regional and global scale.

We removed the ERA-5 results figure.

**RC1.41:** L295ff: From a (soil and hydrologic) physics perspective the main fundament might be that fluxes are driven by gradient depletion and that the degrees of freedom for these fluxes are state dependent (including subscale properties subsumed as hysteresis). The fill-and-spill concept (McDonnell et al. 2021) is a very powerful description of dynamic connectivity and threshold behaviour resulting from the strong non-linearities in soils. However, the depletion of gradients is largely omitted in such models. You might argue (L298f?) that storage-based models do not require an explicit treatment of gradients since it is all implicitly covered by the individual storage and transfer functions. However,

this is not an argument against the importance of soils nor does it solve the standing issue to be capable to convey changing landscape properties into the required storage characteristics.

Please see our response to your general comment on gradient and fluxes.

**RC1.42:** L308ff: Why do you jump from the debate about the concepts back to the debate about available data (which has so far not been really opened)?

We deleted this paragraph.

**RC1.43:** L320ff: Since I read your manuscript as a strong claim for a simplified hydropedologic perceptual model, I find the argument with Occams razor very problematic. I would claim that we are in a situation with plenty of data to challenge our perceptual models and we have the tools to do this (e.g. Höge et al. 2020, Guthke 2017). Occams razor is a perceptual assumption, too.

We don't agree. Firstly, our model is NOT a simplified hydropedologic perceptual model. It regards root zone as an integrated system, rather than simply summarizing isolated parts together, e.g. soil, water, and roots etc. It is controlled by ecosystem's adaptation to climate.

We deleted the argument with Occams razor to make the paper more concise.

**RC1.44:** Again, I sincerely thank the authors for raising this debate. I hope that my review can contribute to sharpening the arguments and to raise awareness about the many aspects that might have fallen a little too short in preparing this manuscript.

We are very grateful to Conrad Jackisch for his very detailed and well-argued comments and for taking ample time to enter in this debate with us. We think and hope that our slightly provocative approach has stimulated the convergence of different viewpoints and hydrological schools.

We have incorporated all the valuable suggestions for improvement and addressed omissions in the literature.

Bibliography

Germann, P.: Viscosity Controls Rapid Infiltration and Drainage, Not the Macropores, Water, 12, 337–15, https://doi.org/10.3390/w12020337, 2020.

Gharari, S., Gupta, H. V., Clark, M. P., Hrachowitz, M., Fenicia, F., Matgen, P., and Savenije, H. H. G.: Understanding the Information Content in the Hierarchy

of Model Development Decisions: Learning From Data, Water Resour Res, 57, https://doi.org/10.1029/2020wr027948, 2021.

Glaser, B., Jackisch, C., Hopp, L., and Klaus, J.: How Meaningful are Plot-Scale Observations and Simulations of Preferential Flow for Catchment Models?, Vadose Zone J, 18, 0–18, https://doi.org/10.2136/vzj2018.08.0146, 2019.

Gupta, H., Clark, M. P., Vrugt, J. A., Abramowitz, G., and Ye, M.: Towards a Comprehensive Assessment of Model Structural Adequacy, Water Resour Res, 48, 1–40, https://doi.org/10.1029/2011wr011044, 2012.

Guthke, A.: Defensible Model Complexity: A Call for Data-Based and Goal-Oriented Model Choice, Groundwater, 55, 646–650, https://doi.org/10.1111/gwat.12554, 2017.

Höge, M., Guthke, A., and Nowak, W.: Bayesian Model Weighting: The Many Faces of Model Averaging, Water-sui, Water, 12, 309, https://doi.org/10.3390/w12020309, 2020.

Jackisch, C., Hassler, S. K., Hohenbrink, T. L., Blume, T., Laudon, H., McMillan, H., Saco, P., and Schaik, L. van: Preface: Linking landscape organisation and hydrological functioning: from hypotheses and observations to concepts, models and understanding, Hydrol Earth Syst Sc, 25, 5277–5285, https://doi.org/10.5194/hess-25-5277-2021, 2021.

Jackisch, C., Knoblauch, S., Blume, T., Zehe, E., and Hassler, S. K.: Estimates of tree root water uptake from soil moisture profile dynamics, Biogeosciences, 17, 5787–5808, https://doi.org/10.5194/bg-17-5787-2020, 2020.

Jackisch, C. and Zehe, E.: Ecohydrological particle model based on representative domains, Hydrol Earth Syst Sc, 22, 3639–3662, https://doi.org/10.5194/hess-22-3639-2018, 2018.

Lange, M., Eisenhauer, N., Sierra, C. A., Beßler, H., Engels, C., Griffiths, R. I., Mellado-Vázquez, P. G., Malik, A. A., Roy, J., Scheu, S., Steinbeiss, S., Thomson, B. C., Trumbore, S. E., and Gleixner, G.: Plant diversity increases soil microbial activity and soil carbon storage, Nature Communications, 6, 1–8, https://doi.org/10.1038/ncomms7707, 2015.

Lin, H., Bouma, J., Pachepsky, Y., Western, A., Thompson, J., Genuchten, R. van, Vogel, H.-J., and Lilly, A.: Hydropedology: Synergistic integration of pedology and hydrology, Water Resour Res, 42, 2509–13, https://doi.org/10.1029/2005wr004085, 2006.

Looy, K. V., Bouma, J., Herbst, M., Koestel, J., Minasny, B., Mishra, U., Montzka, C., Nemes, A., Pachepsky, Y. A., Padarian, J., Schaap, M. G., Tóth, B., Verhoef, A., Vanderborght, J., Ploeg, M. J., Weihermüller, L., Zacharias, S., Zhang, Y., and

Vereecken, H.: Pedotransfer Functions in Earth System Science: Challenges and Perspectives, Rev Geophys, 55, 1199–1256, https://doi.org/10.1002/2017rg000581, 2017.

Loritz, R., Hassler, S. K., Jackisch, C., Allroggen, N., Schaik, L. van, and Wienhöfer, J.: Picturing and modeling catchments by representative hillslopes, 21, 1225–1249, https://doi.org/10.5194/hess-21-1225-2017, 2017.

Loritz, R., Gupta, H., Jackisch, C., Westhoff, M., Kleidon, A., Ehret, U., and Zehe, E.: On the dynamic nature of hydrological similarity, HESS 22, 3663–3684, https://doi.org/10.5194/hess-22-3663-2018, 2018.

McDonnell, J. J., Spence, C., Karran, D. J., Meerveld, H. J. (Ilja) van, and Harman, C. J.: Fill-and-Spill: A Process Description of Runoff Generation at the Scale of the Beholder, Water Resour Res, 57, https://doi.org/10.1029/2020wr027514, 2021.

Melsen, L. A. and Guse, B.: Climate change impacts model parameter sensitivity – implications for calibration strategy and model diagnostic evaluation, Hydrol Earth Syst Sc, 25, 1307–1332, https://doi.org/10.5194/hess-25-1307-2021, 2021.

Novick, K. A., Ficklin, D. L., Baldocchi, D., Davis, K. J., Ghezzehei, T. A., Konings, A. G., MacBean, N., Raoult, N., Scott, R. L., Shi, Y., Sulman, B. N., and Wood, J. D.: Confronting the water potential information gap, Nat Geosci, 15, 158–164, https://doi.org/10.1038/s41561-022-00909-2, 2022.

Ogden, F. L., Allen, M. B., Lai, W., Zhu, J., Seo, M., Douglas, C. C., and Talbot, C. A.: The soil moisture velocity equation, Journal of Advances in Modeling Earth Systems, 9, 1473–1487, https://doi.org/10.1002/2017ms000931, 2017.

Peters, A., Hohenbrink, T. L., Iden, S. C., Genuchten, M. Th. van, and Durner, W.: Prediction of the absolute hydraulic conductivity function from soil water retention data, Hydrology Earth Syst Sci Discuss, 2023, 1–32, https://doi.org/10.5194/hess-2022-431, 2023.

Samaniego, L., Kumar, R., Thober, S., Rakovec, O., Zink, M., Wanders, N., Eisner, S., Schmied, H. M., Sutanudjaja, E. H., Warrach-Sagi, K., and Attinger, S.: Toward seamless hydrologic predictions across spatial scales, Hydrol Earth Syst Sc, 21, 4323–4346, https://doi.org/10.5194/hess-21-4323-2017, 2017.

Vereecken, H., Amelung, W., Bauke, S. L., Bogena, H., Brüggemann, N., Montzka, C., Vanderborght, J., Bechtold, M., Blöschl, G., Carminati, A., Javaux, M., Konings, A. G., Kusche, J., Neuweiler, I., Or, D., Steele-Dunne, S., Verhoef, A., Young, M., and Zhang, Y.: Soil hydrology in the Earth system, Nat Rev Earth Environ, 3, 573–587, https://doi.org/10.1038/s43017-022-00324-6, 2022.

Vogel, H., Gerke, H. H., Mietrach, R., Zahl, R., and Wöhling, T.: Soil hydraulic

conductivity in the state of nonequilibrium, Vadose Zone J, https://doi.org/10.1002/vzj2.20238, 2023.

Wagener, T., Gleeson, T., Coxon, G., Hartmann, A., Howden, N., Pianosi, F., Rahman, M., Rosolem, R., Stein, L., and Woods, R.: On doing hydrology with dragons: Realizing the value of perceptual models and knowledge accumulation, Wiley Interdiscip Rev Water, 8, https://doi.org/10.1002/wat2.1550, 2021.

**Anonymous Referee #2**

This opinion paper makes interesting and bold claims about the importance of soil properties for hydrology. I agree with many of the statements for natural soils and mature ecosystems. However, the majority of our earth is no longer a natural mature ecosystem. We have changed the surface cover drastically and very large areas are under agriculture or are so badly degraded and not in a mature "steady" state that the ecosystem perspective that is advocated in this paper is possibly no longer applicable. I think that this has to be mentioned in the text and that the reader needs to be reminded more frequently that these statements are made for mature natural ecosystems.

Reply: We thank Anonymous Referee #2's endorsement for the scientific significance of our opinion paper, and the agreement on our statements for natural soils and mature ecosystems. We are living in the Anthropocene, but it is still relevant to emphasize the importance of ecosystem understanding. Indeed man has interfered with the ecosystem, but then, they are still ecosystems. Particularly over-year crops and plantations do function as ecosystems. For seasonal crops a different approach needs to be taken, the main difference being that the time scales of adaptation are shorter (rooting depth is limited to the ploughing depth, and crop development is limited by the growing season) and that man interferes in the water balance. But this can be modelled as well and may indeed require soil information. This special treatment of agriculture has been mentioned in the paper. Besides that the world is dramatically affected by agriculture, the main water generating areas, such as hillslopes, mountains, forests and (not overgrazed) grasslands, are not significantly affected by agriculture, even though there is a human footprint.

Further explanation can be found in Line 471-478.

That soil is important is clear in situations where severely degraded ecosystems are restored. It is the restoration of the soil that leads to the very large changes in the flow pathways (from overland flow to subsurface flow) and thus streamflow responses. Indeed, it is the ecosystem that changes the soil properties that lead

to the changes in the hydrological flow pathways and runoff responses, but this does not mean that the soil itself is not important at all. It means that the ecosystem has such a large effect on soil that the ecosystem would be a better predictor to be used in models (because ecosystem and soil properties become correlated as the ecosystem matures and the ecosystem is easier to observe), but it does not mean that soil is not important at all, especially not when one wants to understand processes. I think that some of the statements about soil not being important therefore require a bit more nuance. In particular, the model perspective (rather than process perspective) for some of the claims should be made clearer.

Reply: As the referee will have noticed, the first sentence of the paper is: "Soil is important in hydrology". We do agree with the referee and we shall adjust the paper in other locations if this statement is contradicted elsewhere.

One of the confusing parts of the paper is that the authors state that the rooting zone is important but that soil is not important. This seems to suggest that they think that the rooting zone is not part of the soil. I think that what they mean is that soil texture is not important. To me it seems that most of the time when the authors say that soil is not important, they mean that soil texture is not important. For example when the authors refer to soil in the top down approach of catchment comparisons (Section 3.3), they actually refer to texture, not soil hydraulic properties. I urge the authors to more explicitly state that they focus on the soil texture. A better description of what parts of the soil they think are not important would be really helpful. It will also help if they give their definition of soil early in the paper.

Reply: The referee pointed out that "To me it seems that most of the time when the authors say that soil is not important, they mean that soil texture is not important". We agree that if soil hydraulic properties and moisture conditions were known, it would in principle be possible to determine water movement in the soil. But this approach is impractical if not infeasible due to limitations in data, processes understanding, and computational resources. Soil texture is typically the only soil characteristic that is available at useful scales, which leads to the typical approach of relating soil hydraulic properties to texture. We think that this approach is bound to failure, as soil texture has little to do with soil hydraulic properties at the catchment scale. Our approach relates soil hydraulic properties to the ecosystem, effectively bypassing the need of soil characterization.

Regarding the difference between rootzone and soil, the referee can find our detailed discussion the last paragraph of Section 4.3.

The authors should point out much more clearly (and explicitly) that a major problem is that we use texture in pedotransfer functions to derive the soil characteristics that are related to water flow and storage, especially because

these pedotransfer functions were developed for agricultural soils. The sand or silt content of a soil do not affect water flow or storage. We only attribute such an effect when we use pedotransfer functions to derive properties related to water flow and storage based on the texture. Because the pedotransfer functions were largely derived for agricultural soils, they do not take the effects of structure (and preferential flow) into account.

Reply: We agree completely with this comment. We added Figure 1, showing the procedure of how the pedotransfer function was derived by field sampling and laboratory measurements, and eventually used in soil-based hydrological modelling. The applications of soil-based approach in agricultural studies are discussed in Section 5.1.

The writing of the manuscript could be a bit sharper. At several places, the authors make a good argument for why the ecosystem is important and then conclude that the soil is not important. I think that these sections need to be improved for two reasons. First, reasons are given for why the ecosystem is important but not for why the soil is not important. In particular, no references are given for this second part. In other words, the authors provide arguments for the first part (the ecosystem is important) but not for the second part (soil is not important). Thus either the second part (soil is not important) has to be taken out or arguments and references need to be included for the second part as well. Second, ecosystem and soil are interconnected. It is the ecosystem that changes the soil properties. So one can not directly argue that because the ecosystem is important, the soil is not important. It is still important but the ecosystem is perhaps the better predicting variable to be used in models because it is easier to observe and has a large effect on the soil properties that actually affect how water moves through the soil.

Reply: Thank you for your suggestions. We rewrote many parts of this manuscript to further sharpen our argument.

Other parts of the writing could also be improved. In several sentences words are missing and some other sentences are not clear and should be reformulated. The structure of the paper and individual sections was sometimes unclear to me. For example, section 4.1 consists of four paragraphs. Paragraph one highlights the importance of ET and states that hydrologists focus on discharge instead (but this point was already made on L128). The second paragraph then describes that ecosystems maximize storage and drainage. This section is interesting and fits the caption of this section. One would expect the next paragraph to get deeper into this but the third paragraph describes that the numbers for soil properties used in models don't match the actual measurement values, and the fourth paragraph describes the rebalancing of soil properties that needs to be done in models. While the first two paragraphs sort of fit together and the last two paragraphs as well, the link between the first two and last two is not obvious. It

also means that the second paragraph ends abruptly and this line of thinking could use some more elaboration. In addition, the part on the soil properties and the rebalancing starts abruptly without an introduction. The latter two paragraphs would probably better fit in a separate section on the problematic part of using pedotransfer functions based on texture (see comments above). This is just one example, there are other sections where the flow was unclear and I expect other readers to also wonder how the paragraphs are connected. I made some suggestions in the annotated pdf but there are more places where text could be reordered for a better flow. I don't request that the authors use the suggested order but I do recommend that they carefully read through the paper to see if the order is logical for a reader.

Reply: We followed the referee's suggestions to make our narrative more logical for the readers. The writing has been substantially improved.

Oter specific points:

- L56/139: I think that the problematic part of the use of pedotransfer functions based on texture to derive properties about pores should be described in more detail. Especially knowing that these pedotransfer functions were developed based on cores from agricultural fields and that texture does not really influence the hydraulic conducitivity (e.g., Jarvis et al., 2013; Gupta et al., 2021). See also comments above.

- Reply: Thank you. We added relevant references in the revised MS.

- Section 3.1: I don't think that anyone claims that soil affects the long term water balance more than climate and vegetation. So, I think that it is fine to use this section to highlight that the ecosystem and climate are the main factors that determine the long term water balance but it makes less sense to use this as an argument that soils are not important.

- Reply: The section on long-term water balance is deleted in the revised version.

- L129-132: Yes, land use change (if severe) alters runoff generation, exactly because of the large effect it has on soils. So, I don't think that you can use this argument here to say that soils don't matter. You can use it to make the argument that vegetation has a large effect on the soil properties that actually matter for water flow and storage. Also, it would be good to reference some field studies here (not only model studies).

Reply: We removed the section on long-term water balance.

- L158: But the comparison is basically between a model and a model with more data. I don't think that one should call this observations.

- Reply: We changed the term "observed" by "remote sensing derived".

- L162: But it also mimics the depth to the groundwater – maybe this has a different effect in the two models?

- Reply: Yes, the depth to the groundwater also impacts the evaporation in dry seasons in the Netherlands. The soil-based model heavily relied on detailed soil observations, but did not (sufficiently) consider groundwater replenishment.

- L181: The problem is in part that we use texture here. Texture does not describe the soil pores that are important for storage or flow of water. The problem is that we use pedotransfer functions that are largely based on data from agricultural soils and are not appropriate for forested systems. See also the comments above. Furthermore, soil depth data is usually very rough and not very reliable. Maps of soil properties that actually describe water flow and storage are rarely available. Thus, one could also argue that the big problem is that we don't have soil maps with sufficient information on the properties that actually matter and are related to water flow, and that instead we rely too much on texture and pedotransfer functions.

- Reply: Yes, it is a technical issue to rely too much on soil texture and pedotransfer functions. The more fundamental issue though is whether we understand and model hydrological processes based on the substrate (the soil) or on the active agent (the ecosystems). It is an issue of cause and effect. In the past, we have focused too much on the effect (the soil properties that we observe locally but cannot observe at the relevant scale) instead of trying to understand the agent that creates the soil properties, which acts at the appropriate and observable ecosystem scale.

- L188: I agree that all these processes are intertwined or connected. Therefore, I think that the opinion paper should use more careful wording. It is OK to say that for hydrological modeling it is more useful to look at the ecosystem because the soil properties that matter for hydrology are highly correlated with land cover, and ecosystem properties are much easier to observe or measure. However, if we want to actually understand processes and the factors that affect these processes, it is important to look at the processes. In other words, then we have to look at the partitioning of rainfall into infiltration, overland flow, deeper drainage, etc. and soils are important. I think that this distinction between model application and process understanding should be made more clearly throughout the text.

Reply: Our opinion paper discussed the role of soil in both model development and process understanding. We found that soil is overrated, not only in model

development but also in process understanding. The role of soil is overrated not only in catchment hydrology, but also in hillslope runoff generation under natural condition, land surface evaporation and energy interaction. Even small-scale water movement and pathways are not mainly driven by soil properties but by soil structure, controlled by the ecosystem. Moreover, our argument is about what is the active manager and main driving force, and what is the substrate? What is the dependent variable and what the independent? What is cause, and what is consequence? And eventually what is intuition, and what is realism?

We also believe, and this is hard to prove as yet, that partitioning is controlled by the ecosystem, firstly by interception and throughfall concentration in dripping points where infiltration is facilitated, next by preferential infiltration patterns that are created by biota, and third by subsurface drainage and percolation. From an evolutionary perspective it is reasonable to assume that ecosystems evolve towards survival (a Darwinian hypothesis). This implies that surface runoff is prevented (causing loss of nutrients and fertile topsoil), that depleted moisture stocks are quickly replenished, and that excess water is drained below the root zone (rapid subsurface flow). From a larger ecosystem perspective, one could even go as far as assuming that recharge of groundwater is beneficial to the ecosystem at larger scale, sustaining base flow. We have not even touched upon all the intricacies of how ecosystems manipulate the substrate to its advantage. We have merely shown convincingly (by several model applications) that the root zone storage that ecosystems create are better predictors of hydrological behaviour than soil texture derived storage. In this opinion paper we do not claim that we have all the knowledge required to explain partitioning, we merely point a more promising research direction to untangle hydrological complexity, where the basic assumption is that an active agent, with a clear purpose, creates its own conditions for survival. As a bonus, models based on this approach appear to be simpler, cheaper, less time and resource demanding and better at the job for which they are developed, see for instance Mao and Liu (2019).

Mao, G. and Liu, J.: WAYS v1: a hydrological model for root zone water storage simulation on a global scale, Geosci. Model Dev., 12, 5267–5289, https://doi.org/10.5194/gmd-12-5267-2019, 2019.

- Section 4.2: I am sorry but I don't understand what these ERA5 storage volumes contribute to the arguments of the opinion paper. The volume is one thing, the total flux from repeated filling and emptying is another. Certainly, I agree that the total storage is highest in the root zone but I consider the root zone to be part of the soil. So why is the root zone important but soil not? The paragraph on 277-284 goes some way into explaining this but it could have been added to section 4.1. It would be good if the authors give a definition of soil early in the paper. I have the feeling that often the authors mean soil texture instead of the soil itself.

- Reply: For the relationship between soil and root zone, the Anonymous Referee #2 can find our replies to your main comment in Section 4.3. In this manuscript, when saying soil we mostly mean soil hydraulic properties, including not only soil texture (i.e. sand-, silt-, clay- content), but also porosity, organic matter, and bulk density etc.

- Several minor comments and suggestions are given in the annotated pdf.

Reply: We thank Anonymous Referee #2's comments and suggestions which have been greatly helpful to improve the quality of this manuscript.

References:

Jarvis, N., Koestel, J., Messing, I., Moeys, J., and Lindahl, A.: Influence of soil, land use and climatic factors on the hydraulic conductivity of soil, Hydrol. Earth Syst. Sci., 17, 5185–5195, https://doi.org/10.5194/hess-17-5185-2013, 2013.

Gupta, S., Hengl, T., Lehmann, P., Bonetti, S., and Or, D.: SoilKsatDB: global database of soil saturated hydraulic conductivity measurements for geoscience applications, Earth Syst. Sci. Data, 13, 1593–1612, https://doi.org/10.5194/essd-13-1593-2021, 2021.